# PolyGCL: Graph Contrastive Learning via Learnable Spectral Polynomial Filters

**Jingyu Chen**
Renmin University of China
jy.chen@ruc.edu.cn

**Runlin Lei**
Renmin University of China
runlin_lei@ruc.edu.cn

**Zhewei Wei** [*]
Renmin University of China
zhewei@ruc.edu.cn

## Abstract

Recently, Graph Contrastive Learning (GCL) has achieved significantly superior performance in self-supervised graph representation learning. However, the existing GCL technique has inherent smooth characteristics because of its low-pass GNN encoder and objective based on homophily assumption, which poses a challenge when applied to heterophilic graphs. In supervised learning tasks, spectral GNNs with polynomial approximation excel in both homophilic and heterophilic settings by adaptively fitting graph filters of arbitrary shapes. Yet, their applications in unsupervised learning are rarely explored. Based on the above analysis, a natural question arises: *Can we incorporate the excellent properties of spectral polynomial filters into graph contrastive learning?* In this paper, we address the question by studying the necessity of introducing high-pass information for heterophily from a spectral perspective. We propose PolyGCL, a GCL pipeline that utilizes polynomial filters to achieve contrastive learning between the low-pass and high-pass views. Specifically, PolyGCL utilizes polynomials with learnable filters to generate different spectral views and an objective that incorporates high-pass information through a linear combination. We theoretically prove that PolyGCL outperforms previous GCL paradigms when applied to graphs with varying levels of homophily. We conduct extensive experiments on both synthetic and real-world datasets, which demonstrate the promising performance of PolyGCL on homophilic and heterophilic graphs. Code is available at https://github.com/ChenJY-Count/PolyGCL.

## 1 Introduction

Self-supervised representation learning, which aims to learn informative representations without the demand of costly handcrafted labels, has achieved a wide range of applications in areas such as computer vision, natural language processing, and multimodal (Chen et al., 2020; Grill et al., 2020; Devlin et al., 2019; Radford et al., 2021; Gao et al., 2023). On non-Euclidean graph data, Graph Contrastive Learning (GCL), has become a mainstream research direction in self-supervised scenarios, namely learning representations by capturing consistency across different views and optimizing the objective function based on mutual information maximization to distinguish positive and negative examples (Veličković et al., 2019; Hassani & Khasahmadi, 2020; Peng et al., 2020; Zhu et al., 2020b). Generally, most existing GCL methods rely on the homophily assumption, which means nodes connected by edges tend to have similar node representations. These methods adopt a low-pass filter (such as GCN) encoder and the objective which smooths the representations of neighboring nodes. Therefore, the existing GCL methods have shown excellent performance on homophilic graphs.

However, heterophilic graphs are also prevalent in reality because of the principle of opposites attracting, e.g. dating networks. To address heterophily, many efforts have been made in supervised domain (Pei et al., 2020; Lim et al., 2021). Among them, spectral graph neural networks with learnable polynomial filters adaptively learn appropriate graph filters from graph data in an end-to-end manner (Chien et al., 2021; He et al., 2021; 2022; Guo & Wei, 2023), achieving desirable performance on both homophilic and heterophilic graphs due to more powerful expressiveness. Yet, the applications of spectral GNNs with polynomial filters in self-supervised scenarios are relatively

---

[*]Zhewei Wei is the corresponding author.

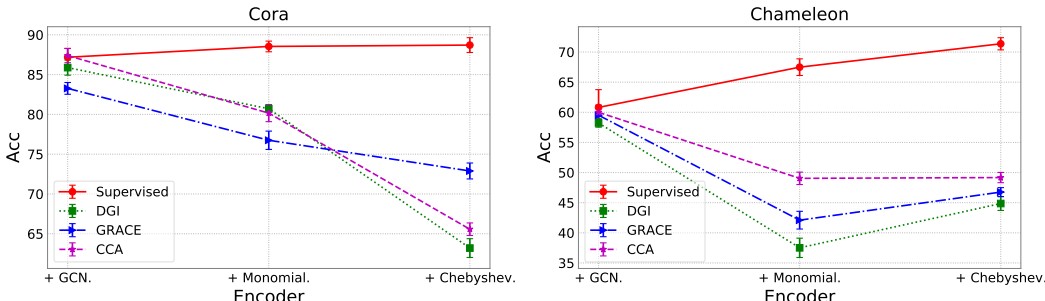

Figure 1: Mean accuracy comparison in the supervised setting (red line) and with 3 self-supervised methods while the encoders are directly substituted with monomial and Chebyshev polynomial filters.

limited, and the traditional paradigm of GCL fails to be applied to heterophilic graphs due to its low-pass nature, leaving self-supervised learning in heterophilic settings an unsolved task. Based on the above, a natural question is: *How can we effectively introduce the properties of spectral polynomial filters into GCL to ensure expressiveness in both homophilic and heterophilic settings?*

To answer the above question, we first consider substituting the low-pass GCN encoder in classic GCL methods with two classic polynomial filters, that are the monomial basis in GPR-GNN (Chien et al., 2021) and the Chebyshev basis in ChebNetII (He et al., 2022). As shown in Figure 1, this simple idea results in performance degradation in self-supervised settings with three different optimization objectives (green, blue, and purple lines), which conflicts with the performance improvement in supervised learning tasks (red line). A similar phenomenon occurs in He et al. (2022) when the spectral polynomial filter is learned in the semi-supervised learning task, demonstrating the difficulty of learning a proper filter without sufficient label information. To address the problem, in this paper, we propose POLYGCL, a novel Graph Contrastive Learning framework via learnable spectral polynomial filters to realize effective learning on graphs with different homophily levels. Specifically, POLYGCL restricts the expressiveness of the polynomial filters from a spectral perspective to construct the low-pass and high-pass views and introduces a simple **linear combination** strategy to construct the optimization objective, which can be theoretically proved to benefit POLYGCL from achieving lower loss in the spectral domain and boosting the performance of downstream tasks. Our contributions can be summarized as follows:

- We propose POLYGCL, which introduces the superior properties of polynomial filters into Graph Contrastive Learning. POLYGCL achieves effective learning on both homophilic and heterophilic graphs without the complex data augmentation or pre-processing in traditional GCL paradigms.
- We theoretically prove the necessity of the high-pass information in heterophilic settings. We also verify that the learning objective constructed by a simple linear combination strategy of the low-pass and high-pass information enjoys theoretical guarantees on downstream tasks.
- Extensive experiments on real-world and synthetic datasets across different homophily levels are conducted to verify the superior performance of POLYGCL without introducing extra complexity. Additional ablation study further confirms our theoretical results.

## 2 RELATED WORK

**Graph Contrastive Learning (GCL).** As a mainstream research topic in self-supervised learning, GCL can be divided into two categories: (1) **Augmentation-based** methods adopt different types of data augmentations to generate different views, and the optimization of the loss function is based on maximizing mutual information between them (Hassani & Khasahmadi, 2020; You et al., 2020; Zhang et al., 2021; Zhu et al., 2020b; 2021b; Liu et al., 2023). We provide a detailed comparison of the augmentation techniques and the space complexity of these methods in Table 7 of Appendix D. (2) **Augmentation-free** methods aim to remove the complex data augmentation or negative sampling strategy but consider inputting the same graph into different encoders to obtain different views and push together the representations of the same node/class from different views (Peng et al., 2020; Mo et al., 2022; Xiao et al., 2022). However, the works mentioned above utilize the low-pass GNN encoders and optimization objectives that smooth neighbor representations inherently, resulting in their unsatisfactory performance on heterophilic graphs.

**Spectral-based GNNs.** From a spectral domain perspective, GCN as a first-order approximation of ChebNet has been proven to be a typical low-pass filter and has been widely used in graph representation learning tasks. Existing works often consider using polynomials to approximate the filter function, which preserves strong fitting capabilities and avoids the $O(N^3)$ complexity of Laplacian eigendecomposition. There are different polynomial choices while taking into account various excellent properties of the basis, such as the monotonic basis of GPR-GNN (Chien et al., 2021), the Chebyshev basis of ChebNet (Defferrard et al., 2016; He et al., 2022), and the non-negative Bernstein basis in BernNet (He et al., 2021), etc. Although spectral polynomial filters have been proven to be capable of fitting arbitrary filter functions that work for both homophilic and heterophilic graphs, there is still a lack of self-supervised applications for them.

## 3 PRELIMINARY

**Problem Formulation.** Given an undirected graph $\mathcal{G} = (\mathcal{V}, \mathcal{E})$, Let $N = |\mathcal{V}|$ and $E = |\mathcal{E}|$ be the number of nodes and edges of the graph, and $F$ denote the feature dimension. We denote $\mathbf{X} \in \mathbb{R}^{N \times F}, \mathbf{A} \in \{0, 1\}^{N \times N}$ as node features and adjacency matrix respectively. In self-supervised node representation learning, the objective is to learn an encoder, $\mathcal{E} : \mathbb{R}^{N \times F} \times \mathbb{R}^{N \times N} \to \mathbb{R}^{N \times D}$, such that $\mathcal{E}(\mathbf{X}, \mathbf{A}) = \mathbf{Z} = \{\mathbf{z}_1, \mathbf{z}_2, \ldots, \mathbf{z}_N\}$ represents high-level representations $\mathbf{z}_i \in \mathbb{R}^D$ for each node $v_i$. The representations may then be used for downstream tasks, such as node classification (Zhu et al., 2020b) and clustering (Bhattacharjee & Mitra, 2021; Yuan et al., 2024).

**Homophily.** Homophily describes the tendency of nodes in the graph to form edges with nodes with the same label. Recently a series of evaluation metrics of graph homophily have been proposed from different perspectives (Lim et al., 2021; Pei et al., 2020). We use the edge homophily degree $h = \frac{|\{(v_i, v_j) : (v_i, v_j) \in \mathcal{E} \wedge y_i = y_j\}|}{E}$ (Zhu et al., 2020a) as the evaluation metric in this work, where the value range is $[0, 1]$. If $h$ approaches 1, the homophily degree of the graph is higher. Otherwise, $h$ approaching 0 indicates a higher degree of heterophily. In Table 6 of Appendix C.2, we list the homophily degree $h$ of some real-world graph datasets involved in this work.

**Spectrum and Graph Filtering.** Define Graph Laplacian as $\mathbf{L} = \mathbf{D} - \mathbf{A}$ and the normalized version as $\tilde{\mathbf{L}} = \mathbf{I} - \mathbf{D}^{-1/2} \mathbf{A} \mathbf{D}^{-1/2}$, where $\mathbf{D}$ is a diagonal matrix with $\mathbf{D}_{ii} = \sum_{i=1}^{N} \mathbf{A}_{ij}$. Decompose the normalized Laplacian as $\tilde{\mathbf{L}} = \mathbf{U} \mathbf{\Lambda} \mathbf{U}^T$, note that $\mathbf{\Lambda} = \text{diag}\{\lambda_0, \ldots, \lambda_{N-1}\}$ is the so-called Laplacian spectrum with eigenvalue $0 = \lambda_0 \leq \lambda_1 \leq \ldots \leq \lambda_{N-1} \leq 2$, and $\mathbf{U}$ is a unitary matrix consisting of eigenvectors. Further, the graph filtering operation on $\mathbf{X}$ is defined as $\mathbf{U} g(\mathbf{\Lambda}) \mathbf{U}^T \mathbf{X}$, where $g(\mathbf{\Lambda})$ denotes the graph filter function. Recent studies (Chien et al., 2021; He et al., 2021; 2022) suggest using polynomials to approximate $g(\mathbf{\Lambda})$ by $K$-order truncation can fit $g(\mathbf{\Lambda})$ of any shape and avoid $O(N^3)$ complexity of eigendecomposition.

## 4 PROPOSED METHOD: POLYGCL

In this section, we revisit the problem of graph filtering in a self-supervised manner. Existing GCL methods mainly focus on the homophilic setting and generally fail when faced with heterophilic graphs. Spectral graph neural networks, designed to handle graphs of varying homophily levels, are primarily constrained to supervised learning tasks. Therefore, we are looking for an approach that enjoys the desirable properties of spectral methods and handles heterophily as well in self-supervised graph representation learning tasks.

The key idea of current spectral methods lies in adaptively learning polynomial filters via supervised signals, e.g., labels. To transfer them into self-supervised scenarios, a natural idea emerges: directly use the learnable graph filter as the encoder and switch the objective function with a self-supervised one. In Figure 1, we implement this idea by swapping the GNN encoder module in three classic self-supervised GCL methods (DGI (Veličković et al., 2019) with the binary cross-entropy loss, GRACE (Zhu et al., 2020b) with the InfoNCE loss and CCA-SSG (Zhang et al., 2021) with the CCA loss inspired from Canonical Correlation Analysis) with two polynomial filters (monomial and Chebyshev basis) on homophilic graph `Cora` and heterophilic graph `Chameleon`. Unfortunately, despite the exceptional performance of GPR-GNN (Chien et al., 2021) (monomial basis) and Cheb-NetII (He et al., 2022) (Chebyshev basis) over GCN in supervised tasks, directly plugging them into self-supervised settings as the encoder causes performance degradation compared with the simple

low-pass GCN. This phenomenon could be attributed to the inconsistency between the powerful fitting ability of polynomial filters and the lack of supervision signals in self-supervised learning scenarios, which further inspires us to add constraints to the shape of polynomial filters, thereby facilitating encoder learning in label-scarce scenarios. As the solution, we present POLYGCL, which learns the polynomial filter based on the fixed low-pass and high-pass channels to construct corresponding views of different spectral properties, thus facilitating the construction of the optimization objective that captures important information in the spectral domain. The implementation of the encoder and the optimization objective are given in Section 4.1 and Section 4.2, respectively.

## 4.1 ENCODER: INTRODUCE THE POLYNOMIAL FILTERS

While existing GCL methods focus on low-pass encoders for homophilic tasks, recent studies find that high-frequency information is essential for heterophilic graphs (He et al., 2022; Lei et al., 2022). To ease the learning process in the self-supervised setting, we decouple the low-pass and high-pass channels of the polynomial filter and restrict it to fit only low-pass and high-pass filter functions. Following He et al. (2022), we adopt Chebyshev polynomials with interpolation as base polynomials, which can be formulated as $\sum_{k=0}^{K} w_k T_k(\hat{\mathbf{L}})\mathbf{X}$, where $\hat{\mathbf{L}} = 2\tilde{\mathbf{L}}/\lambda_{max} - \mathbf{I}$, and $w_k$ is reparameterized as equation 1:

$$w_k = \frac{2}{K+1} \sum_{j=0}^{K} \gamma_j T_k(x_j),\tag{1}$$

where $x_j = \cos\left(\frac{j+1/2}{K+1}\pi\right), j = 0, \ldots, K$ denote the Chebyshev nodes for $T_{K+1}$, and the filter value $h(x_j)$ at the Chebyshev node $x_j$ is reparameterized as a learnable parameter $\gamma_j$ for $x_j \in [-1, 1]$ (Gil et al., 2007; He et al., 2022). Suppose the filter function is non-negative following He et al. (2021), we use the **prefix sum** to make the non-negative learnable parameter $\gamma_j$ increment with $j$ to model the high-pass filter. Likewise, a low pass filter can be reparameterized with **prefix difference** so that the filter value $h(\hat{\lambda})$ decreases in $\hat{\lambda} \in [-1, 1]$. Formally, we have:

$$\gamma_i^H = \sum_{j=0}^{i} \gamma_j, \quad \gamma_i^L = \gamma_0 - \sum_{j=1}^{i} \gamma_j, i = 1, ..., K,\tag{2}$$

where $\gamma_0^H = \gamma_0^L = \gamma_0$. In this way, we have $\gamma_i^H \leq \gamma_{i+1}^H, \gamma_i^L \geq \gamma_{i+1}^L$ (which denotes the filter value $h(x_j)$) for $i = 0, ..., K - 1$, thus guarantee the high pass/low-pass property for $h(\hat{\lambda})$. Based on the above analysis, the low-pass and high-pass polynomial filter encoders can be expressed as equation 3.

$$\mathbf{Z}_L = f_\theta\left(\sum_{k=0}^{K} w_k^L T_k(\hat{\mathbf{L}})\mathbf{X}\right), \quad \mathbf{Z}_H = f_\theta\left(\sum_{k=0}^{K} w_k^H T_k(\hat{\mathbf{L}})\mathbf{X}\right),\tag{3}$$

where $w_k^L$ and $w_k^H$ are obtained via substituting $\gamma$ in equation 1 with $\gamma^L$ and $\gamma^H$ in equation 2, $f_\theta(\cdot)$ represents a shared MLP with parameter $\theta$ for fixed $D$ dimensional output.

## 4.2 OPTIMIZATION OBJECTIVE IN POLYGCL

In POLYGCL, the optimization objective serves as the self-supervision signal for model training. As shown in equation 3, the embedding output by the low-pass encoder $\mathbf{Z}_L$ and the high-pass encoder $\mathbf{Z}_H$ can be considered as low-pass/high-pass spectral views respectively. Thus, an intuitive idea is to obtain the final embedding via linear combination as $\mathbf{Z} = \alpha\mathbf{Z}_L + \beta\mathbf{Z}_H$, where $\alpha, \beta$ are linear coefficients which can also be set as learnable parameters.

We follow Veličković et al. (2019) and Hassani & Khasahmadi (2020) to perform contrastive learning between local patches (node embeddings) and global summaries (graph embeddings), aiming to achieve mutual information maximization between the spectral views. Specifically, we randomly shuffle $\mathbf{X}$ to get the negative low-pass/high-pass embeddings $\tilde{\mathbf{Z}}_L$ and $\tilde{\mathbf{Z}}_H$, and the global summary can be obtained by mean pooling as $\mathbf{g} = \mathbf{Mean}(\mathbf{Z}) = \frac{1}{N}\sum_{i=1}^{N} \mathbf{Z}_i$, where $\mathbf{Z}_i$ denotes the embedding vector for node $v_i$. To score the agreement between node and graph representations, the discriminator $\mathcal{D}$ is defined as $\mathcal{D}(\mathbf{Z}_i, \mathbf{g}) = \sigma(\mathbf{Z}_i\mathbf{W}\mathbf{g}^\top) \in (0, 1)$, where $\mathbf{W} \in \mathbb{R}^{D \times D}$ is the weight matrix and $\sigma$ serves as the sigmoid activation function. Based on the above components, we derive the overall Binary Cross-Entropy (BCE) loss as equation 4:

$$\mathcal{L}_{\text{BCE}} = \frac{1}{4N}\left(\sum_{i=1}^{N} \log\mathcal{D}(\mathbf{Z}_L^i, \mathbf{g}) + \log\left(1 - \mathcal{D}(\tilde{\mathbf{Z}}_L^i, \mathbf{g})\right) + \log\mathcal{D}(\mathbf{Z}_H^i, \mathbf{g}) + \log\left(1 - \mathcal{D}(\tilde{\mathbf{Z}}_H^i, \mathbf{g})\right)\right).\tag{4}$$

Note that we maximize equation 4 for optimization. The whole training paradigm of POLYGCL is presented in Algorithm 1 and Figure 2, coupled with the polynomial encoders shown in equation 3.

---

**Algorithm 1:** Training Algorithm for POLYGCL

---

**Input:** Node features $\mathbf{X}$, input adjacency $\mathbf{A}$, initialized encoders $\mathcal{E}$, initialized coefficients $\alpha, \beta$, maximum iterations $T$, polynomial order $K$.

1  **for** *epoch* $= 0, 1, \ldots, T$ **do**
2      $\tilde{\mathbf{X}} \leftarrow$ shuffle($\mathbf{X}$); % corruption
3      $\gamma_0^L = \gamma_0^H = \gamma_0$ % initialize $\gamma_0^L, \gamma_0^H$ with $\gamma_0$ in $\mathcal{E}$
4      **for** $i = 1, \ldots, K$ **do**
5          $\gamma_i^H = \sum_{j=0}^{i} \mathrm{ReLU}(\gamma_j)$; % high-pass encoder
6          $\gamma_i^L = \gamma_0 - \sum_{j=1}^{i} \mathrm{ReLU}(\gamma_j)$;% low-pass encoder
7      Obtain $\mathcal{E}^L$ and $\mathcal{E}^H$ via $\gamma_i^L$ and $\gamma_i^H$ respectively shown in equation 3
8      $\mathbf{Z}_L \leftarrow \mathcal{E}^L(\mathbf{X}, \mathbf{A}), \mathbf{Z}_H \leftarrow \mathcal{E}^H(\mathbf{X}, \mathbf{A})$; % positive embeddings
9      $\tilde{\mathbf{Z}}_L \leftarrow \mathcal{E}^L(\tilde{\mathbf{X}}, \mathbf{A}), \tilde{\mathbf{Z}}_H \leftarrow \mathcal{E}^H(\tilde{\mathbf{X}}, \mathbf{A})$; % negative embeddings
10     $\mathbf{Z} = \alpha \mathbf{Z}_L + \beta \mathbf{Z}_H$; % linear combination
11     Compute loss via equation 4 and update parameters in $\mathcal{E}^L$ and $\mathcal{E}^H$;

**Output:** $\mathcal{E}^L$ and $\mathcal{E}^H$ with frozen parameters; learned coefficients $\alpha, \beta$.

---

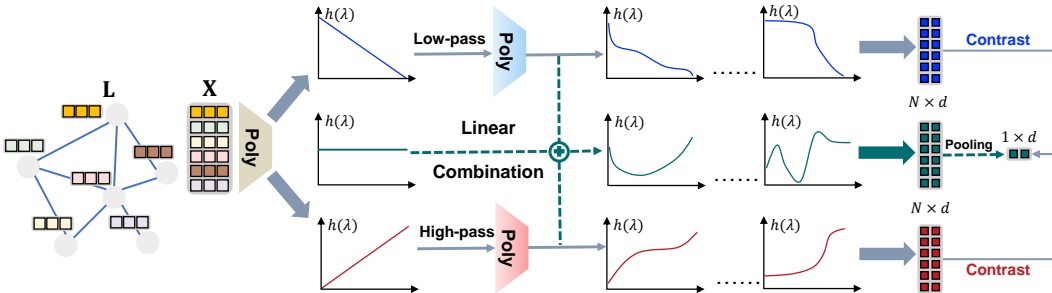

Figure 2: Illustration for the overall pipeline in POLYGCL which takes node features $\mathbf{X}$ and adjacency matrix $\mathbf{A}$ as input. "Poly" and "Init" are short for "polynomial" and "initialization" respectively. There are two components in POLYGCL. The first one is the generation process of low-pass (blue) and high-pass (red) spectral views via polynomial filters. By linear combination, POLYGCL can fit filters of complex shapes shown in green lines. The second one is to perform contrastive learning between the aggregated embedding (green) with the low-pass/high-pass outputs via equation 4.

### 4.3 REVISIT GCL FROM THE SPECTRAL VIEW

In this section, we suppose the downstream task in GCL is binary node classification and the input dimension $F = 1$ to revisit the existing GCL methods from the spectral view. For each node $v_i$, it is attached with a one-hot class label. We denote $\boldsymbol{Y} \in \mathbb{R}^{N \times 2} = (\boldsymbol{y}_0, \boldsymbol{y}_1)$ as the label matrix, where $\boldsymbol{y}_i$ is the indicator vector of class $\mathcal{C}_i, i = 0, 1$. Let the difference of node labels be $\Delta \boldsymbol{y} = \boldsymbol{y}_0 - \boldsymbol{y}_1$. Then we define GCL from the spectral view as two stages: (1) $\mathbf{Z} = \mathbf{U}g(\mathbf{\Lambda})\mathbf{U}^T\mathbf{X}$ (2) $\sigma(\mathrm{MLP}(\mathbf{Z}))$, where $\sigma$ serves as the activation function. Note that only stage (2) involves label $\boldsymbol{y}$. However, as MLP can be considered as all-pass filtering, the graph filtering operation in GCL can be simplified as $\mathbf{Z} = \mathbf{U}g(\mathbf{\Lambda})\mathbf{U}^T\mathbf{X}$. Ideal filtering results in a distinguishable node representation associated with $\Delta \boldsymbol{y}$ to identify node labels. Let $\boldsymbol{\alpha} = \mathbf{U}^\top \Delta \boldsymbol{y}$ and $\boldsymbol{\beta} = \mathbf{U}^\top \mathbf{X}$, we introduce Assumption 1 about the basic correlation between them.

**Assumption 1.** *Assume that $\boldsymbol{\alpha}$ and $\boldsymbol{\beta}$ are positively correlated in the spectral domain, that is, $\mathbb{E}[\boldsymbol{\alpha}] = w\boldsymbol{\beta}, w > 0$.*

Note that cSBM (Chien et al., 2021) graph generation process is in accord with Assumption 1, and further justification is discussed in Appendix C.1.2. We utilize the Spectral Regression Loss (SRL) in EvenNet (Lei et al., 2022) as the evaluation metric of the spectral filter, which is formulated as

$$L(\mathcal{G}) = \sum_{i=0}^{N-1} \left( \frac{\alpha_i}{\sqrt{N}} - \frac{g(\lambda_i)\beta_i}{\sqrt{\sum_{j=0}^{N-1} g(\lambda_j)^2 \beta_j^2}} \right)^2.$$ A graph filter that achieves **lower** SRL is of **higher** performance in the downstream task. Further, Corollary 1 formally demonstrates that there are different filter monotonicity tendencies for homophilic and heterophilic graphs.

**Corollary 1.** *In the task of binary node classification on $k$-regular graph $\mathcal{G}$, for the homophilic graph of $h \to 1$, choose the low-pass $g(\lambda)$ ensures a lower SRL upper bound. Similarly, for the heterophilic graph of $h \to 0$, the high-pass $g(\lambda)$ corresponds to a lower SRL upper bound.*

Given the fact that the low-pass information benefits graphs with high homophily degree and high-pass frequency promotes the learning of graphs with heterophily, a natural idea is to *linearly combine the low-pass and high-pass filtering* together. Theorem 1 provides a special case to illustrate this.

**Theorem 1.** *For a binary node classification task on a $k$-regular graph $\mathcal{G}$, suppose $\lambda_{N-1} = 2$ and we consider the linear bounded filter function, the low-pass filter $g_{low} = c - \frac{c}{2}\lambda \in [0, c]$ and the high-pass filter as $g_{high} = \frac{c}{2}\lambda \in [0, c]$, then a linear combination of the low-pass and high-pass filter $g_{joint} = x g_{low} + y g_{high}, x \geq 0, y \geq 0, x + y = 1$ achieves a lower expected SRL upper bound than $g_{low}$ in heterophilic settings, that is, $\mathbb{E}_x[\hat{L}_{joint}] \leq \mathbb{E}_h[\hat{L}_{low}]$ for $x \sim U(0, 1), h \sim U(\frac{1}{2}, 1)$, where $\hat{L}$ denotes the upper bound for $L$.*

**Remark.** Theorem 1 reveals that in heterophilic graphs, the linear combination strategy of low-pass and high-pass information has a smaller expected SRL upper bound than utilizing the low-pass information only, which provides a theoretical advantage to introduce the high-pass information for modeling heterophilic graphs.

## 4.4 THEORETICAL ANALYSIS

In this section, we explain the effectiveness of POLYGCL from the view of Mutual Information (MI) theory. We first present Proposition 1, which builds a connection with DGI (Veličković et al., 2019):

**Proposition 1.** *The upper bound of $\mathcal{L}_{BCE}$ has the same form as the DGI loss, which indicates $\mathcal{L}_{BCE}$ can be considered as the lower bound of $\mathcal{L}_{DGI}$.*

Proposition 1 claims that maximizing $\mathcal{L}_{\text{BCE}}$ for optimization is equal to maximizing the lower bound of $\mathcal{L}_{\text{DGI}}$. Based on Proposition 1, we can derive Theorem 2.

**Theorem 2.** *Given two deterministic encoder functions $\mathcal{E}^L(\cdot)$ and $\mathcal{E}^H(\cdot)$, which are the low-pass polynomial filter and high-pass polynomial filter respectively. Let $\mathbf{Z}_i^{(k)} = \{\mathbf{z}_j\}_{j \in n(\mathbf{Z}^{(k)}, i)}$ be the neighborhood of the node $i$ in the $k$-th graph that collectively maps to its high-level features, $\mathbf{h}_i^L = \mathcal{E}^L(\mathbf{Z}_i^{(k)}), \mathbf{h}_i^H = \mathcal{E}^H(\mathbf{Z}_i^{(k)})$, where $n$ is the neighborhood function that returns the set of neighborhood indices of node $i$ for graph $\mathbf{Z}^{(k)}$. Let $\mathbf{h}_i^{agg} = \text{Linear}(\mathbf{h}_i^L, \mathbf{h}_i^H)$. Assume that $|\mathbf{Z}_i| = |\mathbf{Z}| = |\mathbf{g}| \geq |\mathbf{h}_i^{agg}|$. Then, $\mathbf{h}_i^{agg}$ that optimising equation 4 also maximizes $\text{MI}(\mathbf{Z}_i^{(k)}; \mathbf{h}_i^{agg})$.*

Note that in Theorem 2, we denote $\mathbf{h}_i^{agg}$ as our final embedding, which aggregates the $K$-hop neighbor information (corresponding to polynomial filters of order $K$) through both the low-pass encoder and high-pass encoder. Maximizing $\text{MI}(\mathbf{Z}_i^{(k)}; \mathbf{h}_i^{agg})$ implies that the final embedding $\mathbf{h}_i^{agg}$, combining both the low-pass and high-pass information, preserves the maximum correlation with the node's original $K$-hop representations.

**Remark.** We can rewrite equation 4 as: $\mathcal{L}_{\text{BCE}} = 2JS\left(P_L^{pos} \parallel P_L^{neg}\right) + 2JS\left(P_H^{pos} \parallel P_H^{neg}\right) - 4\log 2$, which proves that maximizing $\mathcal{L}_{\text{BCE}}$ not only maximizes $\text{MI}(\mathbf{Z}_i^{(k)}; \mathbf{h}_i^{agg})$, but also maximizes the Jensen-Shannon divergence between the positive and negative distributions in both the low-pass and high-pass views. Detailed discussion is deferred to Appendix A.5 due to space limitation.

**Connection with downstream tasks.** In POLYGCL, we have two self-supervised signals: the low-pass information $\mathbf{z}_L$ and the high-pass information $\mathbf{z}_H$. Based on the mutual information maximization interpretation of equation 4, we essentially combine both low-pass and high-pass information as the self-supervised signal. This involves maximizing the mutual information between the representation $\mathbf{h}$ and the joint distribution $(\mathbf{z}_L, \mathbf{z}_H)$. Furthermore, Corollary 2 demonstrates that our method provides a tighter upper bound on the downstream Bayes error (Tsai et al., 2020; Xiao et al., 2022) compared to using only low-pass or high-pass information. This suggests that

Table 1: Mean node classification accuracy (%) with a 95% confidence interval on cSBM graphs. **Boldface** letters indicate the best results and underlining letters denote the second best results.

| Methods | $\phi=-1$ | $\phi=-0.75$ | $\phi=-0.5$ | $\phi=-0.25$ | $\phi=0$ | $\phi=0.25$ | $\phi=0.5$ | $\phi=0.75$ | $\phi=1$ |
|---|---|---|---|---|---|---|---|---|---|
| DGI | $83.04_{\pm0.92}$ | $93.24_{\pm0.54}$ | $85.75_{\pm0.49}$ | $68.41_{\pm0.94}$ | $59.95_{\pm0.78}$ | $68.70_{\pm0.60}$ | $84.04_{\pm0.61}$ | $91.53_{\pm0.42}$ | $82.68_{\pm0.72}$ |
| MVGRL | $68.80_{\pm1.00}$ | $84.35_{\pm0.78}$ | $78.81_{\pm0.63}$ | $64.14_{\pm1.05}$ | $59.09_{\pm1.15}$ | $70.74_{\pm0.73}$ | $89.91_{\pm0.58}$ | $95.95_{\pm0.37}$ | $89.13_{\pm0.55}$ |
| GGD | $82.90_{\pm0.83}$ | $92.76_{\pm0.63}$ | $85.56_{\pm0.58}$ | $66.63_{\pm0.66}$ | $56.00_{\pm0.51}$ | $67.06_{\pm1.06}$ | $84.22_{\pm0.61}$ | $91.75_{\pm0.45}$ | $83.84_{\pm0.76}$ |
| GMI | $54.47_{\pm0.94}$ | $54.38_{\pm0.71}$ | $50.70_{\pm0.91}$ | $50.41_{\pm0.64}$ | $51.79_{\pm0.39}$ | $59.57_{\pm0.93}$ | $82.28_{\pm0.76}$ | $93.74_{\pm0.46}$ | $96.01_{\pm0.48}$ |
| CCA-SSG | $50.55_{\pm0.75}$ | $52.71_{\pm1.08}$ | $51.21_{\pm0.98}$ | $50.88_{\pm0.85}$ | $51.16_{\pm0.67}$ | $56.33_{\pm0.90}$ | $72.41_{\pm1.20}$ | $90.83_{\pm0.62}$ | $62.03_{\pm0.91}$ |
| BGRL | $49.86_{\pm0.77}$ | $49.47_{\pm0.74}$ | $49.95_{\pm0.90}$ | $50.21_{\pm0.87}$ | $54.58_{\pm0.99}$ | $60.80_{\pm0.56}$ | $70.79_{\pm1.01}$ | $74.46_{\pm0.79}$ | $68.69_{\pm0.96}$ |
| GBT | $57.41_{\pm1.43}$ | $64.99_{\pm0.53}$ | $58.84_{\pm0.80}$ | $51.80_{\pm0.87}$ | $57.55_{\pm0.69}$ | $\textbf{72.62}_{\pm\textbf{0.63}}$ | $\textbf{91.09}_{\pm\textbf{0.37}}$ | $\underline{97.80}_{\pm0.25}$ | $96.03_{\pm0.38}$ |
| GRACE | $\underline{98.74}_{\pm0.28}$ | $\textbf{97.55}_{\pm\textbf{0.17}}$ | $90.06_{\pm0.50}$ | $68.74_{\pm1.01}$ | $56.85_{\pm1.12}$ | $66.70_{\pm0.91}$ | $89.50_{\pm0.60}$ | $97.41_{\pm0.25}$ | $98.78_{\pm0.28}$ |
| GCA | $76.56_{\pm0.92}$ | $85.56_{\pm0.40}$ | $\underline{78.96}_{\pm0.43}$ | $\underline{62.32}_{\pm0.89}$ | $58.01_{\pm1.07}$ | $65.30_{\pm1.15}$ | $77.16_{\pm1.03}$ | $81.38_{\pm0.59}$ | $75.54_{\pm0.76}$ |
| GraphCL | $58.82_{\pm1.06}$ | $57.89_{\pm0.68}$ | $52.91_{\pm0.70}$ | $50.18_{\pm0.59}$ | $51.25_{\pm0.76}$ | $55.11_{\pm0.56}$ | $62.54_{\pm1.13}$ | $65.57_{\pm1.17}$ | $71.31_{\pm1.01}$ |
| GREET | $50.82_{\pm0.67}$ | $58.79_{\pm0.52}$ | $59.91_{\pm0.46}$ | $63.57_{\pm0.76}$ | $65.99_{\pm0.64}$ | $71.04_{\pm0.67}$ | $80.17_{\pm0.50}$ | $83.11_{\pm0.53}$ | $75.93_{\pm1.19}$ |
| POLYGCL | $\textbf{98.84}_{\pm\textbf{0.17}}$ | $94.23_{\pm0.31}$ | $\textbf{90.82}_{\pm\textbf{0.50}}$ | $\textbf{75.43}_{\pm\textbf{0.68}}$ | $\textbf{66.51}_{\pm\textbf{0.69}}$ | $69.43_{\pm0.65}$ | $88.22_{\pm0.72}$ | $\textbf{98.09}_{\pm\textbf{0.29}}$ | $\textbf{99.29}_{\pm\textbf{0.23}}$ |

downstream tasks can benefit from the learned representations obtained through our objective function. We attribute this advantage to our utilization of both low-pass and high-pass information as the self-supervised signal. All the detailed proofs are presented in Appendix A.

**Corollary 2.** *Suppose that downstream label* $\mathbf{y}$ *is a M-categorical random variable and the downstream Bayes error on learned representation* $\mathbf{h}$ *as* $P_{\mathbf{h}}^e = \mathbb{E}_{\mathbf{h}}\left[1 - \max_{y\in\mathbf{y}} P(\hat{\mathbf{y}} = y|\mathbf{v})\right]$, *where* $\hat{\mathbf{y}}$ *is the estimation for label from downstream classifier. Then, we have an inequality on the error upper bound* $\sup\left(P_{\mathbf{h}_{agg}}^e\right) \le \min\left(\sup\left(P_{\mathbf{h}_{low}}^e\right), \sup\left(P_{\mathbf{h}_{high}}^e\right)\right)$, *where* $\sup\left(P_{\mathbf{h}}^e\right)$ *denotes the error upper bound for representation* $\mathbf{h}$. *The error upper bound of the low-pass and high-pass representations,* $\mathbf{h}_{low}$ *and* $\mathbf{h}_{high}$, *are denoted as* $\sup\left(P_{\mathbf{h}_{low}}^e\right)$ *and* $\sup\left(P_{\mathbf{h}_{high}}^e\right)$ *respectively.*

## 5 EXPERIMENTS

In this section, we conduct experiments about self-supervised node classification on both synthetic datasets and real-world datasets to evaluate the performance of POLYGCL and gain further insights.

### 5.1 BASELINES AND SETTINGS

We compare our method with three types of GCL baselines as follows based on their optimization objectives. (1) BCE-based GCL methods: DGI (Veličković et al., 2019), MVGRL (Hassani & Khasahmadi, 2020), GMI (Peng et al., 2020) and GGD (Zheng et al., 2022). (2) InfoNCE-based GCL methods: GraphCL (You et al., 2020), GRACE (Zhu et al., 2020b), GCA (Zhu et al., 2021b), GREET (Liu et al., 2023). (3) Invariance-keeping GCL methods: BGRL (Thakoor et al., 2022), GBT (Bielak et al., 2022), CCA-SSG (Zhang et al., 2021). Details about the model architectures and hyperparameters are listed in Appendix F.

**Evaluation Protocol.** We follow the linear evaluation scheme as introduced in Veličković et al. (2019), which can be treated as "two-stage" learning. For the first stage, node features and graph structure without any label information are inputted, and each model is trained in a self-supervised manner. Then at stage 2 (MLP stage), the representations output by the GNN encoder in stage 1 are fixed and used to train, validate, and test via a simple linear classifier. As for the train/valid/test splits on all datasets, we follow Chien et al. (2021) to randomly split the nodes into 60%, 20%, and 20%, and all methods share the same 10 random splits, output embedding size $D$ and hyper-parameters in stage 2 for a fair comparison. See Appendix F.3 to learn more about the detailed settings.

### 5.2 EVALUATION ON SYNTHETIC DATASETS

**Datasets.** To better validate our theoretical results, we adopt the cSBM model to generate graphs with arbitrary homophily degrees following Chien et al. (2021). Note that the homophily degree of cSBM graphs is determined by the parameter $\phi \in [-1, 1]$, where the closer $\phi$ approaches 1, the more homophilic the graph is and vice versa. Details about the cSBM dataset are included in Appendix C.1.

**Results.** The results are shown in Table 1. First, we observe that in self-supervised learning, most models reach their performance peak when $|\phi|$ approaches 1, and the model performance is generally poor when $|\phi|$ approaches 0, which aligns with the findings of supervised learning in

Table 2: Mean node classification accuracy (%) on real-world graphs. **Boldface** letters indicate the best results and underlining letters denote the second best results.

| Methods | Cora | Citeseer | Pubmed | Cornell | Texas | Wisconsin | Actor | Chameleon | Squirrel |
|---|---|---|---|---|---|---|---|---|---|
| DGI | $85.88_{\pm 0.95}$ | $76.44_{\pm 0.80}$ | $82.13_{\pm 0.24}$ | $70.82_{\pm 7.21}$ | $81.48_{\pm 2.79}$ | $75.00_{\pm 2.00}$ | $32.09_{\pm 1.18}$ | $58.23_{\pm 0.70}$ | $38.80_{\pm 0.76}$ |
| MVGRL | $87.36_{\pm 0.64}$ | $78.70_{\pm 0.64}$ | $86.30_{\pm 0.23}$ | $67.70_{\pm 4.75}$ | $73.11_{\pm 4.75}$ | $74.25_{\pm 4.13}$ | $32.98_{\pm 0.53}$ | $57.75_{\pm 1.20}$ | $40.25_{\pm 1.14}$ |
| GGD | $87.21_{\pm 1.08}$ | $79.25_{\pm 0.72}$ | $\underline{85.38}_{\pm 0.25}$ | $80.33_{\pm 1.80}$ | $82.62_{\pm 3.11}$ | $73.25_{\pm 2.25}$ | $32.27_{\pm 1.11}$ | $57.64_{\pm 1.16}$ | $40.87_{\pm 0.66}$ |
| GMI | $85.09_{\pm 1.13}$ | $76.38_{\pm 0.70}$ | $83.06_{\pm 0.24}$ | $62.79_{\pm 7.54}$ | $68.03_{\pm 4.10}$ | $62.13_{\pm 2.88}$ | $32.37_{\pm 1.01}$ | $62.47_{\pm 1.55}$ | $39.82_{\pm 0.93}$ |
| CCA-SSG | $\underline{87.39}_{\pm 0.89}$ | $79.60_{\pm 0.71}$ | $84.96_{\pm 0.20}$ | $78.69_{\pm 3.44}$ | $\underline{87.87}_{\pm 1.64}$ | $82.88_{\pm 1.50}$ | $34.86_{\pm 0.56}$ | $60.00_{\pm 1.20}$ | $41.50_{\pm 0.72}$ |
| BGRL | $84.45_{\pm 0.66}$ | $74.84_{\pm 1.04}$ | $83.06_{\pm 0.29}$ | $59.84_{\pm 2.95}$ | $69.84_{\pm 3.61}$ | $62.88_{\pm 4.13}$ | $32.48_{\pm 0.67}$ | $64.09_{\pm 1.27}$ | $47.02_{\pm 0.88}$ |
| GBT | $84.89_{\pm 1.13}$ | $76.59_{\pm 0.68}$ | $86.10_{\pm 0.23}$ | $59.18_{\pm 9.34}$ | $72.79_{\pm 6.56}$ | $62.38_{\pm 3.00}$ | $34.34_{\pm 0.67}$ | $\underline{68.77}_{\pm 1.25}$ | $\underline{48.86}_{\pm 0.80}$ |
| GRACE | $83.27_{\pm 0.74}$ | $73.79_{\pm 0.60}$ | $81.71_{\pm 0.16}$ | $60.66_{\pm 11.32}$ | $75.74_{\pm 2.95}$ | $72.13_{\pm 2.75}$ | $31.97_{\pm 1.15}$ | $\underline{59.52}_{\pm 1.49}$ | $\underline{42.68}_{\pm 0.90}$ |
| GCA | $84.09_{\pm 0.85}$ | $75.23_{\pm 0.75}$ | $82.01_{\pm 0.31}$ | $53.11_{\pm 9.34}$ | $81.97_{\pm 2.30}$ | $73.50_{\pm 3.00}$ | $31.13_{\pm 0.71}$ | $65.54_{\pm 1.07}$ | $47.13_{\pm 0.61}$ |
| GraphCL | $86.54_{\pm 0.54}$ | $78.99_{\pm 0.50}$ | $85.16_{\pm 0.21}$ | $61.48_{\pm 5.74}$ | $66.07_{\pm 6.07}$ | $60.63_{\pm 3.50}$ | $32.45_{\pm 1.22}$ | $58.49_{\pm 1.31}$ | $42.92_{\pm 0.62}$ |
| GREET | $85.16_{\pm 0.77}$ | $79.06_{\pm 0.44}$ | $85.64_{\pm 0.24}$ | $78.36_{\pm 3.77}$ | $78.03_{\pm 3.94}$ | $\underline{84.63}_{\pm 3.88}$ | $\underline{38.26}_{\pm 0.87}$ | $60.57_{\pm 1.03}$ | $39.76_{\pm 0.74}$ |
| POLYGCL | $\mathbf{87.57}_{\pm 0.62}$ | $\mathbf{79.81}_{\pm 0.85}$ | $\mathbf{87.15}_{\pm 0.27}$ | $\mathbf{82.62}_{\pm 3.11}$ | $\mathbf{88.03}_{\pm 1.80}$ | $\mathbf{85.50}_{\pm 1.88}$ | $\mathbf{41.15}_{\pm 0.88}$ | $\mathbf{71.62}_{\pm 0.96}$ | $\mathbf{56.49}_{\pm 0.72}$ |

Table 3: Experimental results on 5 heterophilic graphs. Accuracy is reported for Roman-empire and Amazon-ratings, and ROC AUC is reported for Minesweeper, Tolokers, and Questions following Platonov et al. (2023). OOM denotes "out of memory".

| Methods | Roman-empire | Amazon-ratings | Minesweeper | Tolokers | Questions |
|---|---|---|---|---|---|
| DGI | $58.57_{\pm 0.26}$ | $42.72_{\pm 0.42}$ | $68.36_{\pm 0.60}$ | $76.29_{\pm 0.66}$ | $74.44_{\pm 0.63}$ |
| MVGRL | $\underline{70.02}_{\pm 0.25}$ | $42.18_{\pm 0.29}$ | $\mathbf{90.07}_{\pm 0.36}$ | $80.86_{\pm 0.63}$ | OOM |
| GGD | $58.04_{\pm 0.40}$ | $43.15_{\pm 0.34}$ | $78.15_{\pm 0.48}$ | $\underline{76.43}_{\pm 0.63}$ | $74.63_{\pm 0.66}$ |
| GMI | $32.33_{\pm 0.27}$ | $40.98_{\pm 0.30}$ | $72.38_{\pm 0.63}$ | $79.89_{\pm 0.62}$ | OOM |
| CCA-SSG | $42.82_{\pm 0.24}$ | $41.23_{\pm 0.25}$ | $72.42_{\pm 0.60}$ | $75.46_{\pm 0.75}$ | $74.64_{\pm 0.57}$ |
| BGRL | $39.34_{\pm 0.32}$ | $41.17_{\pm 0.25}$ | $72.82_{\pm 0.60}$ | $79.73_{\pm 0.61}$ | $72.27_{\pm 0.55}$ |
| GBT | $45.96_{\pm 0.34}$ | $43.58_{\pm 0.28}$ | $72.39_{\pm 0.56}$ | $75.74_{\pm 0.78}$ | $\mathbf{75.98}_{\pm 0.88}$ |
| GRACE | $59.57_{\pm 0.39}$ | $\underline{43.79}_{\pm 0.28}$ | $68.10_{\pm 0.70}$ | $76.31_{\pm 0.71}$ | $74.34_{\pm 0.71}$ |
| GCA | $59.77_{\pm 0.40}$ | $\underline{42.57}_{\pm 0.17}$ | $68.11_{\pm 0.66}$ | $77.26_{\pm 0.61}$ | $75.09_{\pm 0.57}$ |
| GraphCL | $29.92_{\pm 0.30}$ | $37.81_{\pm 0.14}$ | $82.15_{\pm 0.46}$ | $76.88_{\pm 0.60}$ | $60.51_{\pm 1.45}$ |
| GREET | $72.68_{\pm 0.31}$ | $41.19_{\pm 0.25}$ | $82.71_{\pm 0.51}$ | $80.60_{\pm 0.56}$ | OOM |
| POLYGCL | $\mathbf{72.97}_{\pm 0.25}$ | $\mathbf{44.29}_{\pm 0.43}$ | $\underline{86.11}_{\pm 0.43}$ | $\mathbf{83.73}_{\pm 0.53}$ | $\underline{75.33}_{\pm 0.67}$ |

Chien et al. (2021). Specifically, POLYGCL achieves 7 optimal or sub-optimal results in all the 9 settings, holding the best performance when meeting extreme homophily/heterophily ($|\phi| = 1$) or the structural information is useless ($|\phi| = 0$). On other types of datasets, POLYGCL also has comparable performances. We attribute this to the generalization ability of the low-pass/high-pass linear combination strategy across different levels of homophily.

## 5.3 PERFORMANCE ON REAL-WORLD DATASETS

**Datasets.** We conduct the downstream node classification tasks to evaluate the quality of embeddings on 14 real-world benchmark datasets across different homophily degrees. Among them, Cora, Citeseer, and Pubmed (Sen et al., 2008; Yang et al., 2016) are considered homophilic graphs, while Chameleon, Squirrel from Wikipedia (Rozemberczki et al., 2021), the Actor co-occurrence graph and Cornell, Texas, Wisconsin from WebKB (Pei et al., 2020) are denoted as heterophilic graph datasets. We also conduct experiments on 5 larger heterophilic graphs with different structural properties, which are Roman-empire, Amazon-ratings, Minesweeper, Tolokers and Questions (Platonov et al., 2023).

**Results.** We present the performance on real-world datasets in Table 2 and Table 3. Generally, POLYGCL outperforms all baselines in 12 out of 14 benchmarks and achieves the runner-up performance on the other 2 datasets. Notably, POLYGCL slightly boosts the performance compared to other baselines for homophilic graphs while exhibiting a clear performance gain on graphs of heterophily, especially the 15.6% and 7.6% relative improvement on Squirrel and Actor, respectively. POLYGCL's superior performance indicates the exploitation of low-pass and high-pass information can universally benefit representation learning, which is further consistent with our theoretical results.

## 5.4 ABLATION STUDY

To analyze the effectiveness of introducing spectral polynomial graph filters and the high-pass information, we develop a regularized variant of POLYGCL by setting the linear combination

coefficients $\alpha, \beta$ as $\alpha + \beta = 1, \alpha \geq 0, \beta \geq 0$ to satisfy the condition in Theorem 1. In addition, considering that the generation process of cSBM naturally meets Assumption 1, we repeat the experiments in Section 5.2 with the regularized variant.

We first examine the model's preference for low-pass/high-pass information by checking the value of $\alpha \in [0, 1]$, which controls the proportion of low-pass representation $\mathbf{Z_L}$ in the final representation $\mathbf{Z}$. In Figure 3, we observe that $\alpha$ shows an increasing trend as $\phi$ increases in the range of $[-1, 1]$, indicating that in heterophilic graphs ($\phi < 0$), the low-pass information accounts for less of the overall, whereas we draw the opposite conclusion in homophilic cases ($\phi > 0$). In addition, we draw the learned filters (normalized to $[0, 1]$) of the cSBM datasets with different $\phi$ in Figure 4. For $\phi < 0$, POLYGCL tends to learn increasing functions corresponding to high-pass filters, while low-pass filtering dominates in homophilic settings. When $\phi = 0$, which means no structural information is useful, $\alpha$ approaches $0.5$, and the learned filter function is almost an unchangeable line with the all-pass property. The above discussions illustrate that POLYGCL can indeed learn filters of different shapes on graphs across homophily by introducing high-pass information via polynomial filters.

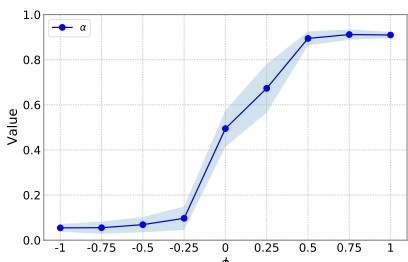
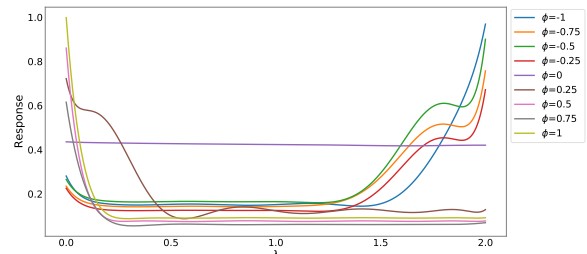

Figure 3: The learned $\alpha$ on the cSBM datasets. The shaded region denotes a 95% confidence interval.

Figure 4: The normalized learned filters of POLYGCL with $\alpha + \beta = 1, \alpha \geq 0, \beta \geq 0$ on cSBM datasets.

## 5.5 COMPLEXITY ANALYSIS

The time complexity of POLYGCL consists of two components: the learning of polynomial filters and the loss computation. Suppose for the graph with $N$ nodes and $E$ edges, the coefficients of the polynomial encoders in POLYGCL can be precomputed in time linear to $K$, therefore the propagation process in $K$-order polynomial filters can be finished in $O(KE)$ time. Meanwhile, the computation of $\mathcal{L}_{BCE}$ costs $O(N)$ time. Thus the overall time complexity of POLYGCL is $O(KE + N)$, which is linear to $K, E$ and $N$. Table 4 shows the results on a large heterophilic graph `arXiv-year` (Lim et al., 2021) with over 1 million edges, where OOM denotes "out of memory" and "-" means failing to finish preprocessing in 24h. POLYGCL still holds the SOTA performance with satisfactory efficiency.

Table 4: Mean classification accuracy on `arXiv-year`.

|  | arXiv-year |
| --- | --- |
| DGI | $40.60_{\pm 0.21}$ |
| GGD | $40.86_{\pm 0.22}$ |
| MVGRL | - |
| BGRL | OOM |
| GBT | $41.90_{\pm 0.26}$ |
| CCA-SSG | $40.76_{\pm 0.25}$ |
| GRACE | OOM |
| POLYGCL | $\mathbf{43.07}_{\pm 0.23}$ |

## 6 CONCLUSION

This paper addresses the problem of dealing with heterophilic issues in self-supervised learning settings. Inspired by the remarkable success of spectral graph neural networks with polynomial approximation in handling heterophily, we seek to extend their desirable properties to the self-supervised domain. We propose POLYGCL, a GCL framework that leverages contrastive learning between the low-pass and high-pass views. Specifically, POLYGCL utilizes the polynomial filters as encoders and incorporates a linear combined objective between low and high frequencies in the spectral domain. Theoretical analysis provides solid evidence that POLYGCL consistently outperforms previous low-pass designs by achieving lower loss. Extensive experiments demonstrate the exceptional performance of POLYGCL across both homophilic and heterophilic settings.

ACKNOWLEDGMENTS

The work was partially done at Gaoling School of Artificial Intelligence, Beijing Key Laboratory of Big Data Management and Analysis Methods, MOE Key Lab of Data Engineering and Knowledge Engineering, and Pazhou Laboratory (Huangpu), Guangzhou, Guangdong 510555, China. This research was supported in part by National Natural Science Foundation of China (No. U2241212, No. 61932001), by Beijing Natural Science Foundation (No. 4222028), by Beijing Outstanding Young Scientist Program No.BJJWZYJH012019100020098, by Alibaba Group through Alibaba Innovative Research Program, and by Huawei-Renmin University joint program on Information Retrieval. We also wish to acknowledge the support provided by the fund for building world-class universities (disciplines) of Renmin University of China, by Engineering Research Center of Next-Generation Intelligent Search and Recommendation, Ministry of Education, Intelligent Social Governance Interdisciplinary Platform, Major Innovation & Planning Interdisciplinary Platform for the "Double-First Class" Initiative, Public Policy and Decision-making Research Lab, and Public Computing Cloud, Renmin University of China.

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

# A  ADDITIONAL PROOFS

## A.1  PROOF OF COROLLARY 1

We first introduce Lemma 1 in Lei et al. (2022).

**Lemma 1.** (Lei et al., 2022) *For a binary node classification task on a $k$-regular graph $\mathcal{G}$, let $h$ be edge homophily and $\lambda_i$ be the $i$-th smallest eigenvalue of $\tilde{\mathbf{L}}$, then*

$$1 - h = \frac{\sum_{i=0}^{N-1} \alpha_i^2 \lambda_i}{2 \sum_i \lambda_i}. \tag{5}$$

*The above equation can be extended to general graphs by replacing the normalized Laplacian $\tilde{\mathbf{L}}$ with the unnormalized $\mathbf{L}$.*

Based on Lemma 1 and Assumption 1, we present the proof of Corollary 1.

**Proof of Corollary 1**

*Proof.* Denote $\boldsymbol{\alpha} = (\alpha_0, \ldots, \alpha_{N-1})^\top, \boldsymbol{\beta} = (\beta_0, \ldots, \beta_{N-1})^\top$. On graph $\mathcal{G}$, note that $L(\mathcal{G}) = \sum_{i=0}^{N-1} \left( \frac{\alpha_i}{\sqrt{N}} - \frac{g(\lambda_i)\beta_i}{\sqrt{\sum_{j=0}^{N-1} g(\lambda_j)^2 \beta_j^2}} \right)^2$. Based on Assumption 1, we suppose each entry of $\boldsymbol{\alpha}$ and $\boldsymbol{\beta}$ are positively correlated in the spectral domain, that is, $\alpha_i = w\beta_i$ for each $i$. Thus, for the low-pass filter $g_{low}$ and the high-pass filter $g_{high}$, the SRLs between the normalized $\boldsymbol{\alpha}$ and $g(\boldsymbol{\lambda})\boldsymbol{\beta}$ are:

$$L_{low}(\mathcal{G}) = 2 - \frac{1}{T_{low}} \left( \sum_{i=0}^{N-1} \alpha_i^2 g(\lambda_i)_{low} \right) \tag{6}$$

$$L_{high}(\mathcal{G}) = 2 - \frac{1}{T_{high}} \left( \sum_{i=0}^{N-1} \alpha_i^2 g(\lambda_i)_{high} \right), \tag{7}$$

where $T_{type} = \frac{\sqrt{N}}{2} \cdot \sqrt{\sum_{i=0}^{N-1} g_{type}(\lambda_i)^2 \alpha_i^2}$. Observing equation 6 and equation 7, we first give the upper bound of $T_{type}$ as inequality 8:

$$T_{type} = \frac{\sqrt{N}}{2} \cdot \sqrt{\sum_{i=0}^{N-1} g_{type}(\lambda_i)^2 \alpha_i^2} \leq c\sqrt{N} \cdot \frac{\sqrt{N}}{2} = \frac{cN}{2}. \tag{8}$$

Note that "$\leq$" in inequality 8 comes from the bounded restriction $g(\lambda) \in [0, c]$. Therefore, there is an upper bound of $L_{low}(\mathcal{G})$ as shown in inequality 9:

$$\begin{aligned} L_{type}(\mathcal{G}) &= 2 - \frac{1}{T_{type}} \left( \sum_{i=0}^{N-1} \alpha_i^2 g(\lambda_i)_{type} \right) \\ &\leq 2 - \frac{2}{cN} \left( \sum_{i=0}^{N-1} \alpha_i^2 g(\lambda_i)_{type} \right) \\ &= 2 - L_t. \end{aligned} \tag{9}$$

Consider minimizing the upper bound of $L_{type}$, that is, just maximizing $L_t$. Below we will explain that it is necessary to use corresponding low-pass/high-pass filtering functions for graphs with homophily/heterophily. Note that for $L_t = \frac{2}{cN} \left( \sum_{i=0}^{N-1} \alpha_i^2 g(\lambda_i)_{type} \right)$, ignoring the constant coefficient, it is essentially a convex combination of $g(\lambda_i)_{type}$, where $\sum_{i=0}^{N-1} \alpha_i^2 = N$ (See Lemma 4 in Lei et al. (2022)).

- **Case** $h \to 0$. This case corresponds to the strong heterophily situation. According to Lemma 1 and $\sum_{i=0}^{N-1} \lambda_i = \text{tr}(\tilde{\mathbf{L}}) = N$, we have equation 10:

$$\sum_{i=0}^{N-1} \alpha_i^2 \lambda_i = 2N(1 - h). \tag{10}$$

Also coupled with $\lambda_{N-1} \leq 2$, we have inequality 11:

$$
\begin{aligned}
\sum_{i=0}^{N-1} \alpha_i^2 \lambda_i &\leq \sum_{i=0}^{N-1} \alpha_i^2 \lambda_{N-1} \\
&= \lambda_{N-1} \left( \sum_{i=0}^{N-1} \alpha_i^2 \right) \\
&\leq 2N.
\end{aligned}
\tag{11}
$$

Therefore $\sum_{i=0}^{N-1} \alpha_i^2 \lambda_i$ has an upper bound $2N$. When $h \to 0$, the value is close to the upper bound. At this time, the corresponding situation is exactly that the larger $\lambda_i$ is allocated more $\alpha_i^2$, that is, $\lambda_i$ and $\alpha_i^2$ have the same monotonicity (both are non-decreasing).

The above proves that $\alpha_i^2$ has a non-decreasing monotonicity in the case of heterophily. In order to maximize $L_t$ (the convex combination of $g(\lambda_i)_{type}$), a larger $g(\lambda_i)_{type}$ value should be assigned to larger $\alpha_i^2$, so it is reasonable for the filter function $g$ to be a high-pass filter $g(\lambda_i)_{high}$.

- **Case** $h \to 1$. The idea is similar to **Case** $h \to 0$, but the process is opposite (note that $\sum_{i=0}^{N-1} \alpha_i^2 \lambda_i \to 0$ at this time). It can be deduced that for homophily graphs, the filter function $g$ is expected to be a low-pass filter $g(\lambda_i)_{low}$.

Based on the analysis of the above two cases, we complete the proof. $\square$

## A.2 PROOF OF THEOREM 1

*Proof.* Below we consider proving that the linear combination of low-pass filtering and high-pass filtering helps to achieve lower SRL.

First, consider the linear combination of low-pass and high-pass filter functions as equation 12:

$$
g(\lambda_i)_{joint} = x g(\lambda_i)_{low} + y g(\lambda_i)_{high},
\tag{12}
$$

where $x > 0, y > 0, x + y = 1$ are linear weighting coefficients. Then the corresponding SRL is defined as equation 13:

$$
\begin{aligned}
L_{joint}(\mathcal{G}) &= 2 - \frac{1}{T_{joint}} \left( \sum_{i=0}^{N-1} \alpha_i^2 g(\lambda_i)_{joint} \right) \\
&= 2 - \frac{1}{T_{joint}} \left( \sum_{i=0}^{N-1} \alpha_i^2 \left( x g(\lambda_i)_{low} + y g(\lambda_i)_{high} \right) \right).
\end{aligned}
\tag{13}
$$

where $T_{joint} = \frac{\sqrt{N}}{2} \cdot \sqrt{\sum_{i=0}^{N-1} \left( x g(\lambda_i)_{low} + y g(\lambda_i)_{high} \right)^2 \alpha_i^2}$.

According to equation 9, the upper bound of SRL $L(\mathcal{G})$ is $\hat{L}(\mathcal{G}) = 2 - L_t$, so the expectation of the upper bound of SRL is determined by $\mathbb{E}[L_t]$, ignoring the constant coefficients in $L_t$, and for the random variable $\alpha_i^2 g(\lambda_i)$, we consider the expectation form of equation 14 for the linear combination filter function $g_{joint}$ (where the variable $x$ obeys a uniform distribution, i.e., $x \sim U(0, 1)$):

$$
\begin{aligned}
\mathbb{E}_x[\alpha^2 g(\lambda)_{joint}] &= \mathbb{E}_x[\alpha^2 \left( x g(\lambda)_{low} + y g(\lambda)_{high} \right)] \\
&= \mathbb{E}_x \left[ (2x - 1)\mathbb{E}[\alpha^2 g(\lambda)_{low}] + c(1 - x) \right] \\
&= \int_{x=0}^{1} (2x - 1)\mathbb{E}[\alpha^2 g(\lambda)_{low}] + c(1 - x) \mathrm{d}x \\
&= \frac{c}{2}.
\end{aligned}
\tag{14}
$$

In addition, for the low-pass filter $g_{low}$, consider its expected form as equation 15:

$$
\begin{aligned}
\mathbb{E}[\alpha^2 g(\lambda)_{low}] &= \mathbb{E}[\alpha^2(c - \frac{c}{2}\lambda)] \\
&= c - \frac{c}{2}\mathbb{E}[\alpha^2\lambda] \\
&= c - \frac{c}{2}2(1 - h) \\
&= ch.
\end{aligned}
\tag{15}
$$

Note that the second "=" to the third "=" in equation 15 applies the conclusion of **Lemma** 1 , namely $\mathbb{E}[\alpha^2\lambda] = 2(1 - h)$. For the heterophilic graph with $h \sim U(\frac{1}{2}, 1)$, that is, $h$ is uniformly distributed on $[\frac{1}{2}, 1]$, we have inequality 16 as follows:

$$
\begin{aligned}
\mathbb{E}_x[\alpha^2 g(\lambda)_{joint}] &= \frac{c}{2} \\
&= \int_{h=\frac{1}{2}}^{1} c\,\mathrm{dh} \\
&\geq \int_{h=\frac{1}{2}}^{1} ch\,\mathrm{dh} \\
&= \mathbb{E}_h[hc] \\
&= \mathbb{E}_h[\alpha^2 g(\lambda)_{low}].
\end{aligned}
\tag{16}
$$

Since the expected upper bound of SRL $\mathbb{E}[\hat{L}] = 2 - \mathbb{E}[L_t] = 2 - \frac{2}{c}\mathbb{E}[\alpha^2 g(\lambda)]$, Therefore in heterophilic settings where $h \sim U(\frac{1}{2}, 1)$, the linear combination of low-pass and high-pass filters guarantees a lower expected SRL upper bound compared with utilizing the low-pass information only, that is, $\mathbb{E}_x[\hat{L}_{joint}] \leq \mathbb{E}_h[\hat{L}_{low}], x \sim U(0, 1), h \sim U(\frac{1}{2}, 1)$, the proof is completed. $\square$

**Additional results:** Above we consider the expected SRL upper bound for the heterophilic graphs, which calculates $\mathbb{E}[\hat{L}]$ when $x \sim U(0, 1), h \sim U(\frac{1}{2}, 1)$. Generally, we assume the homophily degree $h$ is uniformly distributed in $[0, 1]$. Similar to equation 15, the expectation form of high-pass filter $g_{high}$ can be derived as equation 17.

$$
\begin{aligned}
\mathbb{E}[\alpha^2 g(\lambda)_{high}] &= \mathbb{E}[\alpha^2 \frac{c}{2}\lambda] \\
&= \frac{c}{2}\mathbb{E}[\alpha^2\lambda] \\
&= \frac{c}{2}2(1 - h) \\
&= c(1 - h).
\end{aligned}
\tag{17}
$$

In order to minimize the expected upper bound of SRL $\hat{L}$, we only need to maximize $\mathbb{E}_h[\alpha^2 g(\lambda)]$. Based on equation 17, we can easily draw the following two conclusions:

1. For low-pass filtering, $\max_h \mathbb{E}_h[\alpha^2 g(\lambda)_{low}] = \max_h hc$, when $h \to 1$, the expected upper bound of SRL $\hat{L}$ will be minimized, which exactly reveals that low-pass filtering corresponds to homophilic graph; Likewise, for high-pass filtering $\max_h \mathbb{E}_h[\alpha^2 g(\lambda)_{high}] = \max_h(1 - h)c$, when $h \to 0$, the expected upper bound of SRL $\hat{L}$ will be minimized, verifying that high-pass filtering corresponds to the heterophilic graphs.

2. Considering that $\mathbb{E}_x[\alpha^2 g(\lambda)_{joint}] = \frac{c}{2}$, it is irrelevant to the homophily degree $h$ of the graph itself. In addition, we consider $x \sim U(0, 1)$ in the above derivation. However, in fact, the weighting coefficient $x$ in the linear combination can be set to be a learnable parameter during model training. Both of them reflect that linearly combining low-pass and high-pass filtering enjoys better generalization across different homophily levels with a theoretical guarantee.

## A.3 PROOF OF PROPOSITION 1

In this section, before the proof of Proposition 1, we first present Lemma 2, which is used in DGI (Veličković et al., 2019):

**Lemma 2.** (Veličković et al., 2019) *Let $\{\mathbf{Z}^{(k)}\}_{k=1}^{|\mathbf{Z}|}$ be a set of node representations drawn from an empirical probability distribution of graphs, $p(\overline{\mathbf{Z}})$, with finite number of elements, $|\mathbf{Z}|$, such that $p(\mathbf{Z}^{(k)}) = p(\mathbf{Z}^{(k')}) \, \forall k, k'$. Let $\mathcal{R}(\cdot)$ be a deterministic readout function on graphs and $\mathbf{g}^{(k)} = \mathcal{R}(\mathbf{Z}^{(k)})$ be the summary vector of the $k$-th graph, with marginal distribution $p(\mathbf{g})$. The optimal classifier between the joint distribution $p(\mathbf{Z}, \mathbf{g})$ and the product of marginals $p(\mathbf{Z})p(\mathbf{g})$, assuming class balance, has an error rate upper bounded by $\mathrm{Err}^* = \frac{1}{2} \sum_{k=1}^{|\mathbf{Z}|} p(\mathbf{g}^{(k)})^2$. This upper bound is achieved if $\mathcal{R}$ is injective.*

Then we claim that although our objective considers the high-pass and low-pass components simultaneously, the DGI loss still serves as an upper bound for our objective maximization.

**Proof of Proposition 1**

*Proof.* Formally, we rewrite our BCE loss as follows:

$$
\begin{aligned}
\mathcal{L}_{\mathrm{BCE}} &= \frac{1}{4N} \left( \sum_{i=1}^{N} \log \mathcal{D}\left(\mathbf{Z}_L^i, \mathbf{g}\right) + \log\left(1 - \mathcal{D}(\tilde{\mathbf{Z}}_L^i, \mathbf{g})\right) + \log \mathcal{D}\left(\mathbf{Z}_H^i, \mathbf{g}\right) + \log\left(1 - \mathcal{D}(\tilde{\mathbf{Z}}_H^i, \mathbf{g})\right) \right) \\
&= \frac{1}{4N} \left( \sum_{i=1}^{N} \log \mathcal{D}\left(\mathbf{Z}_L^i, \mathbf{g}\right) \cdot \mathcal{D}\left(\mathbf{Z}_H^i, \mathbf{g}\right) + \log\left(1 - \mathcal{D}(\tilde{\mathbf{Z}}_L^i, \mathbf{g})\right) \cdot \left(1 - \mathcal{D}(\tilde{\mathbf{Z}}_H^i, \mathbf{g})\right) \right) \\
&\le \frac{1}{2N} \left( \sum_{i=1}^{N} \log \hat{\mathcal{D}}\left(\mathbf{Z}_L^i, \mathbf{Z}_H^i, \mathbf{g}\right) + \log\left(1 - \hat{\mathcal{D}}(\tilde{\mathbf{Z}}_L^i, \tilde{\mathbf{Z}}_H^i, \mathbf{g})\right) \right) = \mathcal{L}_{\mathrm{DGI}},
\end{aligned}
$$

where $\hat{\mathcal{D}}\left(\mathbf{Z}_L^i, \mathbf{Z}_H^i, \mathbf{g}\right) = \sqrt{\mathcal{D}\left(\mathbf{Z}_L^i, \mathbf{g}\right) \cdot \mathcal{D}\left(\mathbf{Z}_H^i, \mathbf{g}\right)} \in (0, 1)$, and the above inequality can be deduced from the basic inequality.

Considering that the final embeddings $\mathbf{Z} = \alpha \mathbf{Z}_L + \beta \mathbf{Z}_H,$, as a linear combination of $\mathbf{Z}_L$ and $\mathbf{Z}_H$, we know that $\mathbf{Z}$ can represent $\mathbf{Z}_L$ and $\mathbf{Z}_H$ ($\alpha = 1, \beta = 0$ or $\alpha = 0, \beta = 1$, respectively), then rewrite $\hat{\mathcal{D}}\left(\mathbf{Z}_L^i, \mathbf{Z}_H^i, \mathbf{g}\right)$ as $\hat{\mathcal{D}}\left(\mathbf{Z}, \mathbf{g}\right)$, where $\mathbf{g} = \mathcal{R}\left(\mathbf{Z}\right)$.

Thus we obtain that the upper bound for $\mathcal{L}_{\mathrm{BCE}}$ has the form of:

$$
\frac{1}{2N} \left( \sum_{i=1}^{N} \log \hat{\mathcal{D}}(\mathbf{Z}, \mathbf{g}) + \log\left(1 - \hat{\mathcal{D}}(\mathbf{Z}, \mathbf{g})\right) \right),
$$

which is the same as the objective in DGI. As we want to maximize $\mathcal{L}_{\mathrm{BCE}}$ for optimization, it is quite effective to maximize the DGI lower bound; thus, our objective also follows Lemma 2. $\qquad \square$

## A.4 PROOF OF THEOREM 2

We first introduce **Corollary** 1 and **Theorem** 3 in Veličković et al. (2019):

**Corollary 1.** (Veličković et al., 2019) *From now on, assume that the readout function used, $\mathcal{R}$, is injective. Assume the number of allowable states in the space of $\mathbf{g}$, $|\mathbf{g}|$, is greater than or equal to $|\mathbf{Z}|$. Then, for $\mathbf{g}^\star$, the optimal summary under the classification error of an optimal classifier between the joint and the product of marginals, it holds that $|\mathbf{g}^\star| = |\mathbf{Z}|$.*

**Theorem 3.** (Veličković et al., 2019) *Define $\mathbf{g}^*$ as the optimal summary vector under the classification error of an optimal classifier between $p(\mathbf{Z}, \mathbf{g})$ and $p(\mathbf{Z})p(\mathbf{g})$. $\mathbf{g}^* = \arg\max_{\mathbf{g}} \mathrm{MI}(\mathbf{Z}; \mathbf{g})$, where $\mathrm{MI}$ stands for mutual information.*

Based on Theorem 3, in DGI, they claim that for finite input sets and appropriate deterministic functions, minimizing the classification error in the discriminator $\mathcal{D}(\cdot)$ can be used to maximize the MI between the input and output of $\mathcal{R}(\cdot)$. Further, based on Corollary 1, we obtain Theorem 2.

**Proof of Theorem 2**

*Proof.* Given our assumption of $|\mathbf{Z}_i| = |\mathbf{g}|$, there exists an inverse $\mathbf{Z}_i = \mathcal{R}^{-1}(\mathbf{g})$, and therefore $\mathbf{h}_i^L = \mathcal{E}_\theta^L(\mathcal{R}^{-1}(\mathbf{g})), \mathbf{h}_i^H = \mathcal{E}_\theta^H(\mathcal{R}^{-1}(\mathbf{g}))$, i.e. there exists deterministic functions $(\mathcal{E}_\theta^L \circ \mathcal{R}^{-1})$ mapping $\mathbf{g}$ to $\mathbf{h}_i^L$ and $(\mathcal{E}_\theta^H \circ \mathcal{R}^{-1})$ mapping $\mathbf{g}$ to $\mathbf{h}_i^H$.

Considering that $\mathbf{h}_i^{agg} = \text{Linear}(\mathbf{h}_i^L, \mathbf{h}_i^H)$, based on the fact that the linear combination is an invertible function, it is easy to derive that there exists a deterministic function $\text{Linear}\left(\mathcal{E}_\theta^L \circ \mathcal{R}^{-1}, \mathcal{E}_\theta^H \circ \mathcal{R}^{-1}\right)$ mapping $\mathbf{g}$ to $\mathbf{h}_i^{agg}$.

The optimal classifier between the joint distribution $p(\mathbf{h}_i^{agg}, \mathbf{g})$ and the product of marginals $p(\mathbf{h}_i^{agg})p(\mathbf{g})$ then, by Lemma 2, has an error rate upper bound of $\text{Err}^* = \frac{1}{2}\sum_{k=1}^{|\mathbf{Z}|} p(\mathbf{h}_i^{agg})^2$. Therefore, as stated in **Corollary** 1, for the optimal $\mathbf{h}_i^{agg}$, $|\mathbf{h}_i^{agg}| = |\mathbf{Z}_i|$. This result, following the same arguments as in Theorem 3, maximizes the mutual information between the neighborhood and high-level features, $\text{MI}(\mathbf{Z}_i^{(k)}; \mathbf{h}_i^{agg})$. $\qquad\square$

### A.5 CONNECTING EQUATION 4 WITH JS DIVERGENCE

In this section, we further prove that maximize $\mathcal{L}_{\text{BCE}}$ not only maximize $\text{MI}(\mathbf{Z}_i^{(k)}; \mathbf{h}_i^{agg})$ shown in Theorem 2, but also maximize the Jensen-Shannon divergence between the positive and negative distributions in both low pass and high pass views. The following proof is inspired by the theoretical proof in Goodfellow et al. (2014) and Zheng et al. (2022).

*Proof.* Note that for the summary vector, we have:

$$\mathbf{g} = \mathbb{E}(\mathbf{Z}) = \alpha\mathbb{E}(\mathbf{Z}_L) + \beta\mathbb{E}(\mathbf{Z}_H) = \alpha\bar{\mathbf{z}}_L + \beta\bar{\mathbf{z}}_H,$$

which implies that $\mathbf{g}$ is fixed when considering the deterministic encoders in terms of the expectations of certain distributions, namely $P_L$ and $P_H$. Therefore, we can disregard $\mathbf{g}$ to streamline the subsequent derivation.

During training, we maximize to optimize the following objective in equation 18:

$$
\begin{aligned}
\mathcal{L}_{\text{BCE}} &= \mathbb{E}_{\mathbf{z}_L \sim P_L^{pos}} \log\left(\mathcal{D}\left(\mathbf{z}_L\right)\right) + \mathbb{E}_{\mathbf{z}_L \sim P_L^{neg}} \log\left(1 - \mathcal{D}\left(\mathbf{z}_L\right)\right) \\
&\quad + \mathbb{E}_{\mathbf{z}_H \sim P_H^{pos}} \log\left(\mathcal{D}\left(\mathbf{z}_H\right)\right) + \mathbb{E}_{\mathbf{z}_H \sim P_H^{neg}} \log\left(1 - \mathcal{D}\left(\mathbf{z}_H\right)\right), \\
&= \int_{\mathbf{z}_L} P_L^{pos}(\mathbf{z}_L) \log\left(\mathcal{D}\left(\mathbf{z}_L\right)\right) \mathrm{d}\mathbf{z}_L + \int_{\mathbf{z}_L} P_L^{neg}(\mathbf{z}_L) \log\left(1 - \mathcal{D}\left(\mathbf{z}_L\right)\right) \mathrm{d}\mathbf{z}_L \\
&\quad + \int_{\mathbf{z}_H} P_H^{pos}(\mathbf{z}_H) \log\left(\mathcal{D}\left(\mathbf{z}_H\right)\right) \mathrm{d}\mathbf{z}_H + \int_{\mathbf{z}_H} P_H^{neg}(\mathbf{z}_H) \log\left(1 - \mathcal{D}\left(\mathbf{z}_H\right)\right) \mathrm{d}\mathbf{z}_H,
\end{aligned}
\tag{18}
$$

where $P_L^{pos}$ and $P_H^{pos}$ are the distributions of positive embeddings for the low-pass and high-pass encoder respectively, $P_L^{neg}$ and $P_H^{neg}$ are the distributions of negative embeddings. As our objective here is to maximize $\mathcal{L}_{\text{BCE}}$, and $P_{pos}(\mathbf{z}) > 0; P_{neg}(\mathbf{z}) > 0$, we can obtain the optimal solution for $\mathcal{D}(\mathbf{z})$ is $\frac{P_{pos}(\mathbf{z})}{P_{pos}(\mathbf{z}) + P_{neg}(\mathbf{z})}$. This is because for any $(a, b) \in \mathbb{R}^2 \setminus \{0, 0\}$, the maximum of the function $y = a\log(x) + b\log(1 - x)$ is achieved at $\frac{a}{a+b}$ (Goodfellow et al., 2014). By replacing $\mathcal{D}(\mathbf{z})$ with $\frac{P_{pos}(\mathbf{z})}{P_{pos}(\mathbf{z}) + P_{neg}(\mathbf{z})}$ in equation 18, we obtain equation 19:

$$
\begin{aligned}
\mathcal{L}_{\text{BCE}} &= \mathbb{E}_{\mathbf{z}_L \sim P_L^{pos}} \log\left(\frac{P_L^{pos}(\mathbf{z}_L)}{P_L^{pos}(\mathbf{z}_L) + P_L^{neg}(\mathbf{z}_L)}\right) + \mathbb{E}_{\mathbf{z}_L \sim P_L^{neg}} \log\left(\frac{P_L^{neg}(\mathbf{z}_L)}{P_L^{pos}(\mathbf{z}_L) + P_L^{neg}(\mathbf{z}_L)}\right) \\
&\quad + \mathbb{E}_{\mathbf{z}_H \sim P_H^{pos}} \log\left(\frac{P_H^{pos}(\mathbf{z}_H)}{P_H^{pos}(\mathbf{z}_H) + P_H^{neg}(\mathbf{z}_H)}\right) + \mathbb{E}_{\mathbf{z}_H \sim P_H^{neg}} \log\left(\frac{P_H^{neg}(\mathbf{z}_H)}{P_H^{pos}(\mathbf{z}_H) + P_H^{neg}(\mathbf{z}_H)}\right).
\end{aligned}
\tag{19}
$$

From equation 19, we can see it looks similar to the Jensen-Shannon divergence between two distributions $P_1$ and $P_2$, defined as equation 20:

$$JS(P_1 \parallel P_2) = \frac{1}{2}\mathbb{E}_{\mathbf{h} \sim P_1} log(\frac{\frac{P_1}{P_1 + P_2}}{2}) + \frac{1}{2}\mathbb{E}_{\mathbf{h} \sim P_2} log(\frac{\frac{P_2}{P_1 + P_2}}{2}). \tag{20}$$

Thus, we can plug in equation 20 into equation 19 to obtain equation 21:

$$
\begin{aligned}
\mathcal{L}_{\mathrm{BCE}} &= \mathbb{E}_{\mathbf{z}_L \sim P_L^{pos}} \log \left( \frac{\frac{P_L^{pos}(\mathbf{z}_L)}{P_L^{pos}(\mathbf{z}_L) + P_L^{neg}(\mathbf{z}_L)}}{2} \right) + \mathbb{E}_{\mathbf{z}_L \sim P_L^{neg}} \log \left( \frac{\frac{P_L^{neg}(\mathbf{z}_L)}{P_L^{pos}(\mathbf{z}_L) + P_L^{neg}(\mathbf{z}_L)}}{2} \right) - 2 \log 2 \\
&+ \mathbb{E}_{\mathbf{z}_H \sim P_H^{pos}} \log \left( \frac{\frac{P_H^{pos}(\mathbf{z}_H)}{P_H^{pos}(\mathbf{z}_H) + P_H^{neg}(\mathbf{z}_H)}}{2} \right) + \mathbb{E}_{\mathbf{z}_H \sim P_H^{neg}} \log \left( \frac{\frac{P_H^{neg}(\mathbf{z}_H)}{P_H^{pos}(\mathbf{z}_H) + P_H^{neg}(\mathbf{z}_H)}}{2} \right) - 2 \log 2, \\
&= 2 JS \left( P_L^{pos} \parallel P_L^{neg} \right) + 2 JS \left( P_H^{pos} \parallel P_H^{neg} \right) - 4 \log 2,
\end{aligned}
\tag{21}
$$

where we can see maximizing $\mathcal{L}_{\mathrm{BCE}}$ is the same as maximizing $JS \left( P_L^{pos} \parallel P_L^{neg} \right)$ and $JS \left( P_H^{pos} \parallel P_H^{neg} \right)$ at the same time. Thus, by optimizing $\mathcal{L}_{\mathrm{BCE}}$, $P_{pos}$ and $P_{neg}$ tend to be pulled apart and separated in terms of the low-pass view $P_L$ and high-pass view $P_H$. $\qquad\square$

### A.6 PROOF OF COROLLARY 2

In this section, we first present the connection between the proposed objective and the mutual information maximization, then show that the learned representations by our objective provably enjoy a good downstream performance.

We denote the random variable of the input graph as $\mathcal{G}$ and the downstream label as $\mathbf{y}$. For clarity, we omit subscript $i$ in what follows. In POLYGCL, we have two self-supervised signals: the low-pass information inferred by $\mathbf{z}_L$ and the high-pass information $\mathbf{z}_H$. We can formulate $\mathbf{z}_L, \mathbf{z}_H$ from the local structural perspective, which is captured by the representations of the neighbors $\mathbf{z}_L = \{\mathbf{z}_i^L | v_i \in N(v)\}, \mathbf{z}_H = \{\mathbf{z}_i^H | v_i \in N(v)\}$ of node $v$. Then we interpret our objective in equation 4 from the information maximization perspective (Tsai et al., 2020) shown in Theorem 4:

**Theorem 4.** *Optimizing local and global terms in equation 4 is equivalent to maximizing the mutual information between the representation $\mathbf{h}$ and the joint distribution of low-pass and high-pass representations, denoted as $(\mathbf{z}_L, \mathbf{z}_H)$. Let $I(\cdot ; \cdot)$ be the mutual information. Formally, we have:*

$$
\max \mathcal{L}_{BCE} \Rightarrow \max_{\mathbf{h}} I(\mathbf{h}; \mathbf{z}_H, \mathbf{z}_L).
\tag{22}
$$

*Proof.* As $\mathbf{z} = \mathrm{Linear}(\mathbf{z}_L, \mathbf{z}_H)$, according to Theorem 2 and the data processing inequality (Thomas & Joy, 2006), we have:

$$
I(\mathbf{h}; \mathbf{z}_H, \mathbf{z}_L) \geq I(\mathbf{h}; \mathrm{Linear}(\mathbf{z}_H, \mathbf{z}_L)) = I(\mathbf{h}; \mathbf{z}).
$$

Thus maximize $\mathcal{L}_{\mathrm{BCE}}$ is equivalent to maximize the lower bound of $I(\mathbf{h}; \mathbf{z}_H, \mathbf{z}_L)$, which proves $\max \mathcal{L}_{\mathrm{BCE}} \Rightarrow \max_{\mathbf{h}} I(\mathbf{h}; \mathbf{z}_H, \mathbf{z}_L)$. $\qquad\square$

To prove Corollary 2, we first introduce Lemma 3 and Theorem 5.

**Lemma 3.** *For a representation $\mathbf{h}$ that is obtained with a deterministic encoder $f_\theta$ of input graph $\mathcal{G}$ with enough capacity, we have the data processing Markov chain: $(\mathbf{z}_L, \mathbf{z}_H) \leftrightarrow \mathbf{y} \leftrightarrow \mathcal{G} \rightarrow \mathbf{h}$.*

*Proof.* Since $\mathbf{h} = f_\theta(\mathcal{G})$ is a deterministic function of input graph $\mathcal{G}$, we have the following conditional independence: $(\mathbf{z}_L, \mathbf{z}_H) \perp \mathbf{h} | \mathcal{G}$ and $\mathbf{y} \perp \mathbf{h} | \mathcal{G}$ (Federici et al., 2019), which leads to the data processing Markov chain $(\mathbf{z}_L, \mathbf{z}_H) \leftrightarrow \mathbf{y} \leftrightarrow \mathcal{G} \rightarrow \mathbf{h}$. Thus, the proof is completed. $\qquad\square$

Based on Lemma 3, Theorem 5 reveals why the downstream tasks can benefit from the representations learned by our objective function.

**Theorem 5.** *Let $\mathbf{h}_{\mathrm{agg}} = \arg\max_{\mathbf{h}} I(\mathbf{h}; \mathbf{z}_L, \mathbf{z}_H), \mathbf{h}_{\mathrm{low}} = \arg\max_{\mathbf{h}} I(\mathbf{h}; \mathbf{z}_L),$ and $\mathbf{h}_{\mathrm{high}} = \arg\max_{\mathbf{h}} I(\mathbf{h}; \mathbf{z}_H)$. Formally, we have the following inequalities about the task-relevant information:*

$$
I(\mathcal{G}; \mathbf{y}) = \max_{\mathbf{h}} I(\mathbf{h}; \mathbf{y}) \geq I(\mathbf{h}_{\mathrm{agg}}; \mathbf{y}) \geq \max(I(\mathbf{h}_{\mathrm{low}}; \mathbf{y}), I(\mathbf{h}_{\mathrm{high}}; \mathbf{y})).
\tag{23}
$$

*Proof.* We have the data processing inequality (Thomas & Joy, 2006) as inequality 24:

$$I(\mathbf{z}_L, \mathbf{z}_H; \mathcal{G}) \geq I(\mathbf{z}_L, \mathbf{z}_H; \mathbf{h}), I(\mathbf{z}_L, \mathbf{z}_H; \mathcal{G}; \mathbf{y}) \geq I(\mathbf{z}_L, \mathbf{z}_H; \mathbf{h}; \mathbf{y}), I(\mathcal{G}, \mathbf{y}) \geq I(\mathbf{h}, \mathbf{y}). \tag{24}$$

Since $\mathbf{h}_{\text{agg}} = \arg\max_{\mathbf{h}} I(\mathbf{z}_L, \mathbf{z}_H; \mathbf{h})$, we have: $I(\mathbf{z}_L, \mathbf{z}_H; \mathbf{h}_{\text{agg}}) = I(\mathbf{z}_L, \mathbf{z}_H; \mathcal{G})$ and $I(\mathbf{z}_L, \mathbf{z}_H; \mathbf{h}_{\text{agg}}; \mathbf{y}) = I(\mathbf{z}_L, \mathbf{z}_H; \mathcal{G}; \mathbf{y})$. In addition, as $\mathbf{h}_{\text{agg}}$ is deterministic given $\mathbf{z}_L, \mathbf{z}_H$, we also have, $0 \leq I(\mathbf{h}_{\text{agg}}; \mathbf{y}|\mathbf{z}_L, \mathbf{z}_H) \leq H(\mathbf{h}_{\text{agg}}|\mathbf{z}_L, \mathbf{z}_H) = 0$, where $H(\cdot)$ denotes the entropy. Given the above, we have equation 25:

$$\begin{aligned}
I(\mathbf{h}_{\text{agg}}; \mathbf{y}) &= I(\mathbf{h}_{\text{agg}}; \mathbf{y}; \mathbf{z}_L, \mathbf{z}_H) + I(\mathbf{h}_{\text{agg}}; \mathbf{y}|\mathbf{z}_L, \mathbf{z}_H) \\
&= I(\mathcal{G}; \mathbf{y}; \mathbf{z}_L, \mathbf{z}_H) + I(\mathbf{h}_{\text{agg}}; \mathbf{y}|\mathbf{z}_L, \mathbf{z}_H) \\
&= I(\mathcal{G}; \mathbf{y}; \mathbf{z}_L, \mathbf{z}_H) + 0 \\
&= I(\mathcal{G}; \mathbf{y}) - I(\mathcal{G}; \mathbf{y}|\mathbf{z}_L, \mathbf{z}_H) \\
&= \max_{\mathbf{h}} I(\mathbf{h}; \mathbf{y}) - I(\mathcal{G}; \mathbf{y}|\mathbf{z}_L, \mathbf{z}_H) = I(\mathbf{h}_{\text{sup}}; \mathbf{y}) - I(\mathcal{G}; \mathbf{y}|\mathbf{z}_L, \mathbf{z}_H). \tag{25}
\end{aligned}$$

Thus, the mutual information gap between self-supervised representation $\mathbf{h}_{\text{agg}}$ and supervised representation $\mathbf{h}_{\text{sup}}$ is $I(\mathcal{G}; \mathbf{y}|\mathbf{z}_L, \mathbf{z}_H) \geq 0$. Based on the property of mutual information, we further have inequality 26:

$$I(\mathcal{G}; \mathbf{y}|\mathbf{z}_L) = I(\mathcal{G}; \mathbf{y}; \mathbf{z}_H|\mathbf{z}_L) + I(\mathcal{G}; \mathbf{y}|\mathbf{z}_L, \mathbf{z}_H) \geq I(\mathcal{G}; \mathbf{y}|\mathbf{z}_L, \mathbf{z}_H). \tag{26}$$

Similarly, we have $I(\mathcal{G}; \mathbf{y}|\mathbf{z}_H) \geq I(\mathcal{G}; \mathbf{y}|\mathbf{z}_L, \mathbf{z}_H)$. Coupled with equation 25 and equation 26, we have $I(\mathbf{h}_{\text{agg}}; \mathbf{y}) \geq I(\mathbf{h}_{\text{low}}; \mathbf{y})$ and $I(\mathbf{h}_{\text{agg}}; \mathbf{y}) \geq I(\mathbf{h}_{\text{high}}; \mathbf{y})$, which completes the proof. □

Based on Theorem 5, we further prove Corollary 2.

**Proof of Corollary 2**

*Proof.* Suppose the Bayes error $0 \leq P_{\mathbf{h}}^e \leq 1 - \frac{1}{|M|}$, which means classification on the learned embedding is better than random guessing. We utilize inequalities 27 borrowed from Thomas & Joy (2006) and Tsai et al. (2020):

$$P_{\mathbf{h}_{\text{agg}}}^e \leq -\log(1 - P_{\mathbf{h}_{\text{agg}}}^e) \leq H(\mathbf{y}|\mathbf{h}_{\text{agg}}), H(\mathbf{y}|\mathbf{h}_{\text{sup}}) \leq \log 2 + P_{\mathbf{h}_{\text{sup}}}^e \log M. \tag{27}$$

For $H(\mathbf{y}|\mathbf{h}_{\text{agg}})$ and $H(\mathbf{y}|\mathbf{h}_{\text{sup}})$, we have equation 28:

$$\begin{aligned}
H(\mathbf{y}|\mathbf{h}_{\text{agg}}) &= H(\mathbf{y}) - I(\mathbf{h}_{\text{agg}}; \mathbf{y}) \\
&= H(\mathbf{y}) - I(\mathbf{h}_{\text{sup}}; \mathbf{y}) + I(\mathcal{G}; \mathbf{y}|\mathbf{z}_L, \mathbf{z}_H) \\
&= H(\mathbf{y}|\mathbf{h}_{\text{sup}}) + I(\mathcal{G}; \mathbf{y}|\mathbf{z}_L, \mathbf{z}_H), \tag{28}
\end{aligned}$$

where we use equation 25 in the second equality. Combining inequality 27 and equation 28, we have the error upper bound for $\mathbf{h}_{agg}$ as $\sup(P_{\mathbf{h}_{\text{agg}}}^e)$ in inequality 29:

$$P_{\mathbf{h}_{\text{agg}}}^e \leq \log 2 + P_{\mathbf{h}_{\text{sup}}}^e \cdot \log M + I(\mathcal{G}; \mathbf{y}|\mathbf{z}_L, \mathbf{z}_H) \triangleq \sup\left(P_{\mathbf{h}_{\text{agg}}}^e\right). \tag{29}$$

Further using $I(\mathcal{G}; \mathbf{y} \mid \mathbf{z}_L, \mathbf{z}_H) \leq I(\mathcal{G}; \mathbf{y}|\mathbf{z}_L)$ and $I(\mathcal{G}; \mathbf{y}|\mathbf{z}_L, \mathbf{z}_H) \leq I(\mathcal{G}; \mathbf{y}|\mathbf{z}_H)$ shown in equation 26, we have $\sup\left(P_{\mathbf{h}_{\text{agg}}}^e\right) \leq \min\left(\sup\left(P_{\mathbf{h}_{\text{low}}}^e\right), \sup\left(P_{\mathbf{h}_{\text{high}}}^e\right)\right)$, which completes the proof. □

## B   GENERALIZATION OF THEOREM 1

As for the generalized version of Theorem 1, we first introduce the generalized version of Lemma 1 borrowed from Lei et al. (2022).

**Lemma 4.** *(Lei et al., 2022) Given the **unnormalized graph Laplacian** $\mathbf{L} = \mathbf{U}_L \mathbf{\Lambda}_L \mathbf{U}_L^\top$ and its eigenvalues $\{\lambda_i'\}$, denote the number of edges on the graph is $m$, and label difference as $\Delta \mathbf{y} = \mathbf{y_0} - \mathbf{y_1} \in \mathbb{R}^{N \times 1}$. For the **unnormalized** spectrum of label difference on $\mathbf{L}$, denoted as $\boldsymbol{\alpha}' = U_L^\top \Delta \mathbf{y} = (\alpha_0', \alpha_1', \dots, \alpha_{N-1}')^\top$, we have:*

$$\sum_{i=0}^{N-1} \lambda_i' = 2m, 1 - h = \frac{\sum_{i=0}^{N-1} (\alpha_i')^2 \lambda_i'}{2 \sum_{j=0}^{N-1} \lambda_j'}. \tag{30}$$

Based on Lemma 4, following the proof sketch of Theorem 1, it can be easily proved that Theorem 1 still holds for the unnormalized graph Laplacian $\mathbf{L}$ as a generalized version, shown in Theorem 6.

**Theorem 6.** *(generalized version of Theorem 1 on unnormalized Laplacian $\mathbf{L}$ ) For a binary node classification task on graph $\mathcal{G}$ with $n$ nodes and $m$ edges, we consider the linear bounded filter function and the unnormalized Laplacian $\mathbf{L}$ with $\lambda' \geq 0$, for the low-pass filter $g_{low} = c - \frac{nc}{4m}\lambda'$ and the high-pass filter as $g_{high} = \frac{nc}{4m}\lambda'$, then a linear combination of the low-pass and high-pass filter $g_{joint} = xg_{low} + yg_{high}, x \geq 0, y \geq 0, x + y = 1$ achieves a lower expected SRL upper bound than $g_{low}$ in heterophilic settings, that is, $\mathbb{E}_x[\hat{L}_{joint}] \leq \mathbb{E}_h[\hat{L}_{low}]$ for $x \sim U(0,1), h \sim U(\frac{1}{2}, 1)$, where $\hat{L}$ denotes the upper bound for $L$.*

**Proof of Theorem 6**

*Proof.* According to equation 9, the upper bound of SRL $L(\mathcal{G})$ is $\hat{L}(\mathcal{G}) = 2 - L_t$, so the expectation of the upper bound of SRL is determined by $\mathbb{E}[L_t]$, ignoring the constant coefficients in $L_t$, and for the random variable $\alpha_i^2 g(\lambda_i')$, we consider the expectation form of equation 31 for the linear combination filter function $g_{joint}$ (where the variable $x$ obeys a uniform distribution, i.e., $x \sim U(0,1)$):

$$
\begin{aligned}
\mathbb{E}_x[\alpha^2 g(\lambda')_{joint}] &= \mathbb{E}_x[\alpha^2 \left(xg(\lambda')_{low} + yg(\lambda')_{high}\right)] \\
&= \mathbb{E}_x\left[(2x-1)\mathbb{E}[\alpha^2 g(\lambda')_{low}] + c(1-x)\right] \\
&= \int_{x=0}^1 (2x-1)\mathbb{E}[\alpha^2 g(\lambda')_{low}] + c(1-x)\mathrm{d}x \\
&= \frac{c}{2}.
\end{aligned}
\tag{31}
$$

In addition, for the low-pass filter $g_{low}$, consider its expected form as equation 32:

$$
\begin{aligned}
\mathbb{E}[\alpha^2 g(\lambda')_{low}] &= \mathbb{E}[\alpha^2(c - \frac{nc}{4m}\lambda')] \\
&= c - \frac{nc}{4m}\mathbb{E}[\alpha^2 \lambda'] \\
&= c - \frac{nc}{4m}\frac{4m}{n}(1-h) \\
&= ch.
\end{aligned}
\tag{32}
$$

Note that the second "=" to the third "=" in equation 32 applies the conclusion of Lemma 4, namely $\mathbb{E}[\alpha^2 \lambda'] = \frac{4m}{n}(1-h)$. For the heterophilic graph with $h \sim U(\frac{1}{2}, 1)$, that is, $h$ is uniformly distributed on $[\frac{1}{2}, 1]$, we have inequality 33 as follows:

$$
\begin{aligned}
\mathbb{E}_x[\alpha^2 g(\lambda')_{joint}] &= \frac{c}{2} \\
&= \int_{h=\frac{1}{2}}^1 c\mathrm{d}h \\
&\geq \int_{h=\frac{1}{2}}^1 ch\mathrm{d}h \\
&= \mathbb{E}_h[hc] \\
&= \mathbb{E}_h[\alpha^2 g(\lambda')_{low}]
\end{aligned}
\tag{33}
$$

Since the expected upper bound of SRL $\mathbb{E}[\hat{L}] = 2 - \mathbb{E}[L_t] = 2 - \frac{2}{c}\mathbb{E}[\alpha^2 g(\lambda')]$, Therefore in heterophilic settings where $h \sim U(\frac{1}{2}, 1)$, the linear combination of low-pass and high-pass filters guarantees a lower expected SRL upper bound compared with utilizing the low-pass information only, that is, $\mathbb{E}_x[\hat{L}_{joint}] \leq \mathbb{E}_h[\hat{L}_{low}], x \sim U(0,1), h \sim U(\frac{1}{2}, 1)$, the proof is completed. $\square$

Note that the slight difference between Theorem 1 and Theorem 6 lies in the constant coefficients of the low-pass/high-pass linear functions, which are $\frac{c}{2}$ and $\frac{nc}{4m}$ respectively. In addition, Theorem 6 gets rid of the $k$-regular restriction in Theorem 1 by introducing the unnormalized graph Laplacian $\mathbf{L}$.

As for the binary classification task in Theorem 1, there is a series of works based on spectral analysis of heterophily that adopt the two-class setting(Lei et al., 2022; Chen et al., 2022; Ma et al., 2022;

Luan et al., 2023). Besides, the theoretical analysis in Chen et al. (2022) states that the analysis of multi-class cases can be simplified via the "One vs Others" reduction, that is, for $C$ class classification task, denoting $\boldsymbol{y}_0' = \boldsymbol{y}_0$ and $\boldsymbol{y}_1' = \sum_{l=1}^{C-1} \boldsymbol{y}_l$, thus we can transform the multi-class cases into binary classification.

Based on the above discussion, Theorem 1 makes reasonable simplifications and has the potential to be extended to a general form of graph Laplacian or multi-class classification cases, which demonstrate the necessity of introducing high-pass information in heterophilic settings.

## C    DETAILS OF EXPERIMENTS

### C.1    SYNTHETIC DATASETS

#### C.1.1    GENERATION PROCESS OF CSBM

In order to generate a graph with varying degrees of homophily, we follow Chien et al. (2021) to generate cSBM datasets. Denote a cSBM graph $\mathcal{G}$ as $\mathcal{G} \sim \text{cSBM}(n, f, \lambda, \mu)$, where $n$ is the number of nodes, $f$ is the dimension of features, and $\lambda$ and $\mu$ are hyperparameters controlling the proportion of contributions from the graph structure and node features respectively. We partition nodes into two classes of equal size $(n/2)$ and assign each node an $f$-dimensional feature vector which is randomly sampled from a class-specific Gaussian distribution as:

$$x_i = \sqrt{\frac{\mu}{n}} y_i u + \frac{Z_i}{\sqrt{f}}, \tag{34}$$

where $y_i \in \{-1, +1\}$ denotes the label of node $v_i$, $u \sim N(0, I/f)$ and $Z$ is a random noise term.

Assume the average degree of the generated graph is $d$, and the adjacency matrix $\mathbf{A}$ of the generated cSBM graph is defined as:

$$\mathbb{P}\left[\mathbf{A}_{ij} = 1\right] = \begin{cases} \frac{d + \lambda\sqrt{d}}{n} & \text{if } v_i v_j > 0 \\ \frac{d - \lambda\sqrt{d}}{n} & \text{otherwise.} \end{cases} \tag{35}$$

To control the homophily degree $h$ of the generated graphs, the parameter $\Phi$ is introduced in the form of $\phi = \frac{2}{\pi} \arctan\left(\frac{\lambda\sqrt{\xi}}{\mu}\right)$, where $\xi$ is a control factor defined as $\xi = \frac{n}{f}$. The degree of homophily in the graph is determined by the parameter $\phi \in [-1, 1]$, where a larger $|\phi|$ value indicates that the graph provides stronger topological information. On the other hand, when $\phi = 0$, only node features are informative for prediction. It is noteworthy that the closer $\phi$ approaches 1, the more homophilic the graph is, and conversely, if $\phi < 0$, the graph tends to exhibit heterophily.

To generate informative cSBM graphs, we need to satisfy the condition $\lambda^2 + \frac{\mu^2}{\xi} = 1 + \epsilon$ where $\epsilon > 0$ (Deshpande et al., 2018). In practice, we choose $n = 5000, f = 2000, d = 5, \epsilon = 3.25$ for all graphs. The choices of $\lambda$ and $\mu$ and the resulting homophily ratio $h$ and number of edges $|E|$ are listed in Table 5.

Table 5: Statistics of cSBM datasets.

| $\phi$ | -1 | -0.75 | -0.50 | -0.25 | 0 | 0.25 | 0.50 | 0.75 | 1 |
|---|---|---|---|---|---|---|---|---|---|
| $\lambda$ | -2.06 | -1.90 | -1.46 | -0.79 | 0.00 | 0.79 | 1.46 | 1.90 | 2.06 |
| $\mu$ | 2.00 | 1.25 | 2.30 | 3.01 | 3.26 | 3.01 | 2.30 | 1.25 | 2.00 |
| $|E|$ | 25,058 | 25,134 | 25,390 | 25,140 | 24,872 | 25,024 | 24,866 | 25,382 | 25,038 |
| $h$ | 0.036 | 0.072 | 0.171 | 0.322 | 0.503 | 0.677 | 0.825 | 0.925 | 0.960 |

#### C.1.2    JUSTIFICATION FOR ASSUMPTION 1

**Justification.** The widely used graph generative model, contextual stochastic block model (cSBM) naturally adheres to Assumption 1 while generating graphs with different homophily levels. Specif-

ically, cSBM generates node features and edges independently via equation 34 and equation 35 respectively. As shown in equation 34, it is exactly a linear correlation between node label $y_i$ and feature $x_i$, which also can be extended to the spectral domain by left multiplying $\mathbf{U}^T$. While taking expectations at the same time, we demonstrate that the generation mechanism of cSBM is consistent with Assumption 1 and further reflect the rationality of Assumption 1.

## C.2 REAL-WORLD DATASETS

We introduce the details of the real-world datasets as follows. and the statistics of them are shown in Table 6.

- `Cora`, `Citeseer` and `Pubmed` (Sen et al., 2008) are three citation networks that are considered classic homophilic graphs. In these graphs, nodes represent papers and edges represent citation relationships between two papers. The features consist of bag-of-word representations of the papers, while the labels indicate the research topic of each paper.

- `Cornell`, `Texas` and `Wisconsin` (Pei et al., 2020) are three heterophilic networks originating from the WebKB[1] project, where nodes are web pages of the computer science departments of different universities and edges are hyperlinks between them. The features of each page are represented as bag-of-words, and the labels indicate the types of web pages.

- `Chameleon` and `Squirrel` (Pei et al., 2020) are two heterophilic networks based on Wikipedia. The nodes denote web pages in Wikipedia and edges denote links between them. The features consist of informative nouns in the Wikipedia pages, and labels indicate the average traffic of the web pages.

- `Actor` (Pei et al., 2020) is an actor co-occurrence network where nodes denote actors and edges indicate two actors have co-occurrence in the same movie. The features indicate the keywords in the Wikipedia pages, and the labels are the words of corresponding actors. It is a typical heterophilic graph.

- `Roman-empire`, `Amazon-ratings`, `Minesweeper`, `Tolokers` and `Questions` (Platonov et al., 2023) are 5 large heterophilic graphs with different structural properties and from different fields. They are proposed to ease the problem of existing heterophilic graphs, e.g., there are a large number of duplicate nodes in `Chameleon` and `Squirrel`, leading to training and test data leakage.

- `arXiv-year` (Lim et al., 2021) is the ogbn-arXiv (Hu et al., 2020) network with different labels, that is, set the class labels to be the year that the paper is posted instead of the paper subject area and balance the class ratios approximately. It is considered as a typical large heterophilic graph.

# D BASELINES

We summarize the baseline methods based on their optimization objectives as follows.

- **Binary Cross-Entropy (BCE) loss.** Inspired by Deep Infomax (Hjelm et al., 2019) in computer vision, DGI (Veličković et al., 2019) contrasts the node embeddings with the global summary with a JSD estimator and BCE loss. To extend the idea of DGI, MVGRL (Hassani & Khasahmadi, 2020) introduces multi-view contrastiveness with diffusion augmentation, and GMI (Peng et al., 2020) focuses on a local scope with the first-order neighborhood. Further, GGD (Zheng et al., 2022) simplifies the discriminator in DGI and proposes the group discrimination objective based on BCE loss to achieve effective and efficient learning.

- **InfoNCE loss.** GRACE (Zhu et al., 2020b) performs contrastive learning on two augmented views including the feature masking and edge dropout with an InfoNCE objective. GCA (Zhu et al., 2021b) improves the performance of GRACE by introducing adaptive augmentation techniques to capture the important features and structural information. GraphCL (You et al., 2020) considers the combination of multiple augmentations to diversify the augmented views. GREET (Liu et al., 2023) proposes an edge heterophily discriminating mechanism based

---

[1]`http://www.cs.cmu.edu/afs/cs.cmu.edu/project/theo-11/www/wwkb/`

Table 6: Statistics of real-world datasets.

| Dataset | Num. of nodes | Num. of edges | Features | Classes | Homophily |
|---|---|---|---|---|---|
| Cora | 2,708 | 5,278 | 1,433 | 7 | 0.810 |
| Citeseer | 3,327 | 4,552 | 3,703 | 6 | 0.736 |
| Pubmed | 19,717 | 44,324 | 500 | 3 | 0.802 |
| Cornell | 183 | 298 | 1,703 | 5 | 0.305 |
| Texas | 183 | 325 | 1,703 | 5 | 0.108 |
| Wisconsin | 251 | 515 | 1,703 | 5 | 0.196 |
| Chameleon | 2,277 | 36,101 | 2,277 | 5 | 0.235 |
| Squirrel | 5,201 | 217,073 | 2,089 | 5 | 0.224 |
| Actor | 7,600 | 30,019 | 932 | 5 | 0.219 |
| Roman-empire | 22,622 | 65,854 | 300 | 18 | 0.047 |
| Amazon-ratings | 24,492 | 186,100 | 300 | 5 | 0.380 |
| Minesweeper | 10,000 | 78,804 | 7 | 2 | 0.683 |
| Tolokers | 11,758 | 1,038,000 | 10 | 2 | 0.595 |
| Questions | 48,921 | 307,080 | 301 | 2 | 0.840 |
| arXiv-year | 169,343 | 1,166,243 | 128 | 5 | 0.222 |

on the InfoNCE objective to achieve effective Graph Contrastive Learning on heterophilic graphs.

- **Invariance-keeping loss.** BGRL (Thakoor et al., 2022) adopts a bootstrapping scheme inspired by BYOL (Grill et al., 2020), which only contrasts node embeddings between the output of the online network and the corresponding target network, thus achieving negative-sample free. GBT (Bielak et al., 2022) utilizes a cross-correlation-based loss based on the redundancy-reduction principle to build contrastiveness between embedding dimensions. CCA-SSG (Zhang et al., 2021) further considers optimizing a feature-level objective inspired by classical Canonical Correlation Analysis to capture the invariance between augmented views effectively.

Besides, we can also summarize the existing GCL methods from the augmentation techniques and the space complexity in Table 7.

Table 7: Technical comparison of the representative GCL methods regarding data augmentation and space complexity, where $N, E$, and $D$ denote the number of nodes, edges, and the size of the output embeddings, respectively. Diffusion denotes graph diffusion via Personalized PageRank or heat kernel. Multiple denotes multiple augmentation methods, including edge removing, edge adding, node dropping, and subgraph induced by random walks. "/" indicates that no such augmentation exists. (Here we consider the node shuffling strategy as the corruption technique but not for data augmentation.)

| Method | Topology Aug. | Feature Aug. | Space. |
|---|---|---|---|
| DGI (Veličković et al., 2019) | / | / | $O(N)$ |
| MVGRL (Hassani & Khasahmadi, 2020) | Diffusion | / | $O(N)$ |
| GMI (Peng et al., 2020) | / | / | $O(N + |E|)$ |
| GGD (Zheng et al., 2022) | / | / | $O(N)$ |
| GRACE (Zhu et al., 2020b) | Edge Removing | Feature Masking | $O(N^2)$ |
| GCA (Zhu et al., 2021b) | Edge Removing | Feature Masking | $O(N^2)$ |
| GraphCL (You et al., 2020) | Multiple | Feature Dropout | $O(N^2)$ |
| GREET (Liu et al., 2023) | Edge Removing | Feature Masking | $O(N^2)$ |
| BGRL (Thakoor et al., 2022) | Edge Removing | Feature Masking | $O(N)$ |
| GBT (Bielak et al., 2022) | Edge Removing | Feature Masking | $O(N)$ |
| CCA-SSG (Zhang et al., 2021) | Edge Removing | Feature Masking | $O(D^2)$ |

# E    SUPPLEMENT RELATED WORK

There is a series of works on Graph Contrastive Learning with heterophily. HLCL (Yang & Mirza-soleiman, 2023) employs a fixed set of polynomials (i.e., $\tilde{\mathbf{A}}^k$ and $\tilde{\mathbf{L}}^k$ represent the low-pass/high-pass filters respectively) to generate spectral views for further contrast, while similar ideas are also used in graph clustering (Xie et al., 2023) and supervised settings (Bo et al., 2021). Furthermore, FiGURe (Ek-bote et al., 2023), which focuses on self-supervised learning tasks on heterophilic graphs and involves filter augmentations, performs contrastive learning in different spectral views generated by filter banks. Motivated by the spectral contrastive loss proposed by HaoChen et al. (2021), SP-GCL (Wang et al., 2022) conducts contrastive learning based on the transformed graphs constructed by the output embeddings instead of the original graphs, thus achieving effective learning on graphs of different homophily levels. Additionally, Local-GCL (Zhang et al., 2022) devises a kernelized contrastive loss with linear complexity for GCL, which also shows effectiveness in heterophilic graphs.

There are also GCL methods using spectral augmentations. Ghose et al. (2023) proposes a set of well-motivated graph transformation operations derived via graph spectral analysis, which are spectral graph cropping and graph frequency components reordering. SpCo (Liu et al., 2022) studies the necessity of high-frequency information in GCL and learns the optimal augmentation from the spectral view.

Compared with the above, POLYGCL does not require specially designed or complex preprocessing steps for spectrum augmentations but achieves the contrast between low-pass and high-pass information by directly optimizing the corresponding decoupled filters. In this way, POLYGCL has the ability to learn filters of any shape in self-supervised learning, which ensures its expressiveness to address GCL with heterophily. Besides, these works related to the graph spectrum mainly focused on the analysis of eigenvalues while POLYGCL cares more about the learning of filter functions to adapt to homophilic/heterophilic settings.

In summary, to the best of our knowledge, POLYGCL is the first to achieve efficient learning of low-pass and high-pass filters via polynomial approximation in a self-supervised setting.

# F    IMPLEMENTATION DETAILS

## F.1    EXPERIMENTAL DEVICE

We conduct all the experiments on a machine with an NVIDIA A100 80GB PCIe, Intel Xeon CPU (2.20 GHz) with 40 cores, and 512 GB of RAM.

## F.2    MODEL ARCHITECTURES

We refer to the official code to implement all the baseline models with the help of PyTorch Geomet-ric(Fey & Lenssen, 2019), DGL (Wang et al., 2019) and PyGCL (Zhu et al., 2021a) libraries. The URL and commit number are presented in Table 8.

Table 8: Codes & commit numbers.

|  | URL | Commit |
|---|---|---|
| DGI | https://github.com/PetarV-/DGI | 61baf67 |
| MVGRL | https://github.com/kavehhassani/mvgrl | 628ed2b |
| GMI | https://github.com/zpeng27/GMI | 3491e8c |
| GGD | https://github.com/zyzisastudyreallyhardguy/graph-group-discrimination | 7cf72db |
| GRACE | https://github.com/CRIPAC-DIG/GRACE | 51b4496 |
| GCA | https://github.com/CRIPAC-DIG/GCA | cd6a9f0 |
| GraphCL | https://github.com/Shen-Lab/GraphCL | a0c8c97 |
| GREET | https://github.com/yixinliu233/GREET | 8bcc940 |
| BGRL | https://github.com/nerdslab/bgrl | 60f9f19 |
| GBT | https://github.com/pbielak/graph-barlow-twins | ec62580 |
| CCA-SSG | https://github.com/hengruizhang98/CCA-SSG | cea6e73 |

Table 9: Details of the hyper-parameters tuned by grid search on cSBM datasets.

| $\phi$ | lr | wd | lr1 | wd1 | lr alpha | activation | initial low value | initial high value | range |
|---|---|---|---|---|---|---|---|---|---|
| $-1$ | 1e-2 | 0.0 | 5e-3 | 1e-6 | 1e-2 | ReLU | 2 | 0 | 4 |
| $-0.75$ | 1e-2 | 0.0 | 1e-3 | 0.0 | 1e-2 | ReLU | 2 | 0 | 4 |
| $-0.5$ | 1e-2 | 0.0 | 1e-3 | 0.0 | 1e-2 | ReLU | 2 | 0 | 4 |
| $-0.25$ | 1e-2 | 0.0 | 1e-3 | 0.0 | 1e-2 | ReLU | 2 | 0 | 4 |
| $0$ | 1e-2 | 0.0 | 1e-4 | 0.0 | 5e-3 | PReLU | 4 | 0 | 3 |
| $0.25$ | 1e-2 | 1e-4 | 5e-3 | 0.0 | 5e-3 | ReLU | 4 | 0 | 3 |
| $0.5$ | 1e-2 | 0.0 | 1e-3 | 0.0 | 1e-2 | ReLU | 2 | 0 | 2 |
| $0.75$ | 1e-2 | 0.0 | 1e-3 | 0.0 | 1e-2 | ReLU | 2 | 0 | 2 |
| $1$ | 1e-2 | 0.0 | 1e-3 | 0.0 | 1e-2 | ReLU | 2 | 0 | 2 |

Table 10: Details of the hyper-parameters tuned by grid search on real-world datasets.

| Dataset | lr | wd | lr1 | wd1 | epochs | patience | dprate | dropout | activation | batch norm |
|---|---|---|---|---|---|---|---|---|---|---|
| Cora | 5e-4 | 1e-3 | 2e-3 | 0.0 | 500 | 20 | 0.3 | 0.3 | ReLU | False |
| Citeseer | 1e-4 | 0.0 | 5e-4 | 0.0 | 1,000 | 20 | 0.2 | 0.3 | ReLU | False |
| Pubmed | 1e-4 | 0.0 | 1e-3 | 1e-3 | 1,000 | 20 | 0.6 | 0.0 | PReLU | True |
| Cornell | 1e-4 | 0.0 | 1e-3 | 0.0 | 500 | 20 | 0.8 | 0.5 | ReLU | False |
| Texas | 5e-3 | 0.0 | 1e-3 | 0.0 | 500 | 20 | 0.4 | 0.5 | ReLU | False |
| Wisconsin | 1e-4 | 0.0 | 1e-3 | 0.0 | 500 | 20 | 0.1 | 0.7 | PReLU | True |
| Chameleon | 1e-3 | 0.0 | 1e-3 | 0.0 | 1,000 | 50 | 0.3 | 0.2 | PReLU | True |
| Squirrel | 1e-3 | 0.0 | 1e-3 | 0.0 | 500 | 20 | 0.2 | 0.0 | PReLU | True |
| Actor | 1e-2 | 0.0 | 1e-3 | 0.0 | 500 | 20 | 0.2 | 0.3 | ReLU | False |
| Roman-empire | 1e-4 | 0.0 | 1e-3 | 0.0 | 500 | 20 | 0.8 | 0.5 | ReLU | False |
| Amazon-ratings | 5e-3 | 0.0 | 1e-3 | 0.0 | 500 | 20 | 0.3 | 0.4 | ReLU | False |
| Minesweeper | 1e-4 | 0.0 | 1e-3 | 0.0 | 500 | 20 | 0.4 | 0.1 | ReLU | False |
| Tolokers | 1e-2 | 0.0 | 1e-3 | 0.0 | 500 | 20 | 0.6 | 0.4 | PReLU | True |
| Questions | 1e-3 | 0.0 | 1e-3 | 0.0 | 500 | 20 | 0.1 | 0.2 | ReLU | False |

Table 11: Details of the hyper-parameters tuned by grid search on `arXiv-year`.

| Dataset | lr | wd | lr1 | wd1 | epochs | patience | dprate | dropout | activation | batch norm |
|---|---|---|---|---|---|---|---|---|---|---|
| arXiv-year | 1e-3 | 0 | 1e-3 | 0.0 | 1,000 | 20 | 0.5 | 0.5 | ReLU | False |

### F.3 HYPERPARAMETER SETTINGS.

Note that for the downstream node classification tasks based on the learned embeddings, we set the embedding size for all models as the same and fix the learning rate $l_2 = 0.01$ and weight decay $w_2 = 0.0$ at the second MLP stage for fair comparison.

**Experiments on the synthetic datasets.** On the cSBM datasets, we test the performance of the regularized variant of POLYGCL to satisfy the condition in Theorem 1 with the constraint $\alpha + \beta = 1, \alpha \geq 0, \beta \geq 0$. In our experiments, we utilize the sigmoid activation function to satisfy the above constraint. In addition, for fixed parameters, we set the order of polynomials $K = 10$, the output embedding size $D = 512$, and the early stopping patience in the training process as 20. The other hyperparameters are listed in Table 9, where parameters "initial low value", "initial high value", and "range" are utilized for the initialization of the low-pass/high-pass filters.

**Experiments on the real-world datasets.** On 14 real-world datasets, we choose the order of polynomials $K = 10$ and the output embedding size $D = 512$. Table 10 shows the other parameters tuned by grid search, and it is noted that for Minesweeper, Tolokers, and Questions, we adopt ROC AUC as the evaluation metric following Platonov et al. (2023). Furthermore, in Table 11, we conduct the self-supervised node classification task on `arXiv-year` with 5 fixed 50/25/25 train/val/test splits as introduced in Lim et al. (2021). We choose $D = 256$ for POLYGCL and other baselines in `arXiv-year` to address the OOM issue that arises with most baselines when $D$ is set to 512.

# G    ADDITIONAL EXPERIMENTAL RESULTS

## G.1    ADDITIONAL BASELINES: GCL WITH HETEROPHILY

We include two new baselines, namely SP-GCL (Wang et al., 2022) and HLCL (Yang & Mirza-soleiman, 2023) for further comparison, which are GCLs that address heterophily. We conduct experiments on both synthetic and real-world datasets. The results are summarized in Table 12, Table 13, and Table 14. Specifically, shown in Table 12, although claimed to tackle heterophily, these two methods still suffer from performance drop in extreme heterophilic cSBM settings when $\phi$ approaches $-1$. In contrast, POLYGCL consistently holds superior performance over the two new baselines on both synthetic and real-world datasets.

Table 12: Mean node classification accuracy (%) with a 95% confidence interval on cSBM graphs compared with additional baselines.

| Methods | $\phi = -1$ | $\phi = -0.75$ | $\phi = -0.5$ | $\phi = -0.25$ | $\phi = 0$ | $\phi = 0.25$ | $\phi = 0.5$ | $\phi = 0.75$ | $\phi = 1$ |
|---|---|---|---|---|---|---|---|---|---|
| SP-GCL | $65.82_{\pm 1.03}$ | $73.19_{\pm 0.88}$ | $68.37_{\pm 0.89}$ | $63.72_{\pm 0.68}$ | $59.36_{\pm 0.98}$ | $73.01_{\pm 0.51}$ | $85.52_{\pm 0.67}$ | $94.13_{\pm 0.38}$ | $88.22_{\pm 0.49}$ |
| HLCL | $66.03_{\pm 0.83}$ | $67.66_{\pm 0.59}$ | $70.62_{\pm 0.63}$ | $60.80_{\pm 0.53}$ | $58.92_{\pm 0.87}$ | $65.80_{\pm 0.40}$ | $79.25_{\pm 0.79}$ | $97.12_{\pm 0.82}$ | $93.07_{\pm 0.80}$ |
| POLYGCL | $\mathbf{98.84}_{\pm 0.17}$ | $\mathbf{94.23}_{\pm 0.31}$ | $\mathbf{90.82}_{\pm 0.50}$ | $\mathbf{75.43}_{\pm 0.68}$ | $\mathbf{66.51}_{\pm 0.69}$ | $\underline{69.43}_{\pm 0.65}$ | $\mathbf{88.22}_{\pm 0.72}$ | $\mathbf{98.09}_{\pm 0.29}$ | $\mathbf{99.29}_{\pm 0.23}$ |

Table 13: Mean node classification accuracy (%) on real-world graphs compared with additional baselines.

| Methods | Cora | Citeseer | Pubmed | Cornell | Texas | Wisconsin | Actor | Chameleon | Squirrel |
|---|---|---|---|---|---|---|---|---|---|
| SP-GCL | $82.99_{\pm 1.18}$ | $75.54_{\pm 1.06}$ | $85.74_{\pm 0.21}$ | $69.41_{\pm 1.49}$ | $69.76_{\pm 1.23}$ | $69.34_{\pm 0.77}$ | $35.92_{\pm 0.67}$ | $69.23_{\pm 1.23}$ | $53.05_{\pm 1.05}$ |
| HLCL | $85.53_{\pm 1.03}$ | $76.79_{\pm 0.60}$ | $85.13_{\pm 0.18}$ | $64.00_{\pm 8.98}$ | $78.38_{\pm 5.08}$ | $79.50_{\pm 4.50}$ | $40.56_{\pm 0.70}$ | $63.86_{\pm 1.34}$ | $44.49_{\pm 0.68}$ |
| POLYGCL | $\mathbf{87.57}_{\pm 0.62}$ | $\mathbf{79.81}_{\pm 0.85}$ | $\mathbf{87.15}_{\pm 0.27}$ | $\mathbf{82.62}_{\pm 3.11}$ | $\mathbf{88.03}_{\pm 1.80}$ | $\mathbf{85.50}_{\pm 1.88}$ | $\mathbf{41.15}_{\pm 0.88}$ | $\mathbf{71.62}_{\pm 0.96}$ | $\mathbf{56.49}_{\pm 0.72}$ |

Table 14: Experimental results on 6 heterophilic graphs compared with additional baselines.

| Methods | Roman-empire | Amazon-ratings | Minesweeper | Tolokers | Questions | arXiv-year |
|---|---|---|---|---|---|---|
| SP-GCL | $63.17_{\pm 0.22}$ | $43.11_{\pm 0.32}$ | $81.76_{\pm 0.61}$ | $80.73_{\pm 0.62}$ | $75.08_{\pm 0.49}$ | $42.56_{\pm 0.12}$ |
| HLCL | $67.75_{\pm 0.19}$ | $43.92_{\pm 0.26}$ | $79.34_{\pm 0.59}$ | $78.99_{\pm 0.67}$ | $74.92_{\pm 0.65}$ | OOM |
| POLYGCL | $\mathbf{72.97}_{\pm 0.25}$ | $\mathbf{44.29}_{\pm 0.43}$ | $\mathbf{86.11}_{\pm 0.43}$ | $\mathbf{83.73}_{\pm 0.53}$ | $\mathbf{75.33}_{\pm 0.67}$ | $\mathbf{43.07}_{\pm 0.23}$ |

## G.2    ABLATION STUDY: LINEAR COEFFICIENTS

We include experiments where $\alpha$ and $\beta$ are set to zero separately as the ablation study of linear coefficients. Specifically, $\alpha = 0, \beta = 0$ means only the high-pass/low-pass filter is reserved respectively. The results are given in Table 15.

We observe that the results of POLYGCL ($\alpha = 0$) remain comparable in heterophilic datasets, while POLYGCL ($\beta = 0$) shows better adaptability to homophilic settings. The results reveal that the high-pass information is indispensable for graphs with large heterophily, and so is the low-pass information for homophilic graphs. The results of this ablation study are further consistent with the low-pass/high-pass preference for homophilic/heterophilic settings in Figure 3 in Section 5.4. Note that POLYGCL achieves better performance when optimized over both the parameters $\alpha, \beta$ in most datasets.

## G.3    SUBSTITUTING EQUATION 4 WITH THE NT-XENT LOSS

We consider directly substituting our optimization loss in equation 4 with the NT-Xent loss used in GraphCL (You et al., 2020). Besides, the augmentation strategies and other settings are aligned with GraphCL. The results are listed in Table 16.

We observe that the results of POLYGCL (NT-Xent) are slightly lower than POLYGCL. This phenomenon can be attributed to the introduction of structural augmentations (e.g., edge-dropping or subgraph-sampling) that destroy the spectral properties of the original graph, which is not conducive

Table 15: Mean node classification accuracy (%) on real-world graphs compared with two variants of POLYGCL.

| Methods | Cora | Citeseer | Pubmed | Cornell | Texas | Wisconsin | Actor | Chameleon | Squirrel |
|---|---|---|---|---|---|---|---|---|---|
| POLYGCL | $87.57_{\pm 0.62}$ | $\mathbf{79.81}_{\pm 0.85}$ | $\mathbf{87.15}_{\pm 0.27}$ | $\mathbf{82.62}_{\pm 3.11}$ | $\mathbf{88.03}_{\pm 1.80}$ | $\mathbf{85.50}_{\pm 1.88}$ | $\mathbf{41.15}_{\pm 0.88}$ | $\mathbf{71.62}_{\pm 0.96}$ | $\mathbf{56.49}_{\pm 0.72}$ |
| POLYGCL ($\alpha = 0$) | $70.67_{\pm 0.48}$ | $64.22_{\pm 0.93}$ | $76.41_{\pm 0.35}$ | $80.16_{\pm 2.62}$ | $86.52_{\pm 1.97}$ | $82.07_{\pm 2.75}$ | $38.28_{\pm 0.39}$ | $68.21_{\pm 1.40}$ | $52.10_{\pm 0.80}$ |
| POLYGCL ($\beta = 0$) | $\mathbf{87.65}_{\pm 0.67}$ | $78.76_{\pm 0.75}$ | $86.64_{\pm 0.17}$ | $76.23_{\pm 5.41}$ | $82.56_{\pm 2.13}$ | $68.88_{\pm 2.50}$ | $37.36_{\pm 0.46}$ | $65.08_{\pm 1.27}$ | $48.52_{\pm 0.71}$ |

Table 16: Mean node classification accuracy (%) on real-world graphs as for different losses.

| Methods | Cora | Citeseer | Pubmed | Cornell | Texas | Wisconsin | Actor | Chameleon | Squirrel |
|---|---|---|---|---|---|---|---|---|---|
| POLYGCL | $87.57_{\pm 0.62}$ | $79.81_{\pm 0.85}$ | $87.15_{\pm 0.27}$ | $82.62_{\pm 3.11}$ | $88.03_{\pm 1.80}$ | $85.50_{\pm 1.88}$ | $41.15_{\pm 0.88}$ | $71.62_{\pm 0.96}$ | $56.49_{\pm 0.72}$ |
| POLYGCL (NT-Xent) | $84.40_{\pm 0.93}$ | $76.83_{\pm 0.94}$ | $82.63_{\pm 0.30}$ | $81.48_{\pm 2.46}$ | $84.43_{\pm 2.95}$ | $81.75_{\pm 3.50}$ | $38.95_{\pm 0.81}$ | $69.17_{\pm 0.94}$ | $53.30_{\pm 1.30}$ |

to the learning of spectral filters in POLYGCL. However, POLYGCL (NT-Xent) still holds competitiveness with other baselines across different homophilic and heterophilic datasets in Table 2, which reflects the universality and effectiveness of the POLYGCL framework.

## G.4 DISCUSSIONS ON THE REPARAMETERIZATION IN EQUATION 2

We utilize the reparameterization techniques in equation 2 to ensure the low-pass and high-pass properties of the learned filters during the learning process, which correspond to the filter functions with incremental and decremental values, respectively.

To verify its effectiveness, we list the comparison between the results of POLYGCL and POLYGCL (wo-RP) in Table 17, where POLYGCL (wo-RP) denotes POLYGCL without ReParameterization. By decoupling the low-pass and high-pass information via simple reparameterization, POLYGCL benefits from a more stable model training process and improves the performance on homophilic/heterophilic datasets.

Table 17: Mean node classification accuracy (%) on real-world graphs: Reparameterization Analysis.

| Methods | Cora | Citeseer | Pubmed | Cornell | Texas | Wisconsin | Actor | Chameleon | Squirrel |
|---|---|---|---|---|---|---|---|---|---|
| POLYGCL | $87.57_{\pm 0.62}$ | $79.81_{\pm 0.85}$ | $87.15_{\pm 0.27}$ | $82.62_{\pm 3.11}$ | $88.03_{\pm 1.80}$ | $85.50_{\pm 1.88}$ | $41.15_{\pm 0.88}$ | $71.62_{\pm 0.96}$ | $56.49_{\pm 0.72}$ |
| POLYGCL (wo-RP) | $85.09_{\pm 0.78}$ | $76.74_{\pm 0.80}$ | $84.39_{\pm 0.29}$ | $78.26_{\pm 3.02}$ | $84.11_{\pm 2.29}$ | $79.50_{\pm 3.50}$ | $38.14_{\pm 0.96}$ | $65.98_{\pm 0.95}$ | $50.06_{\pm 1.01}$ |

In addition, the reparameterization technique based on non-negativity and prefix sum/difference can be easily extended to other polynomial bases, which further verifies the robustness of this technique. In detail, as for the Bernstein polynomial (He et al., 2021), which can also learn arbitrary filters, we directly utilize this reparameterization on the coefficients $\theta_k$ which proves to be equivalent to the filter value $h(\lambda)$. Besides, as for the Monomial polynomial in GPR-GNN (Chien et al., 2021), we can also consider non-negative coefficients to ensure the low-pass/high-pass property, for instance, $\sum_{i=0}^{K} \gamma_i (2\mathbf{I} - \tilde{\mathbf{L}})^i$ and $\sum_{i=0}^{K} \gamma_i \tilde{\mathbf{L}}^i$ for low-pass/high-pass filters respectively, where the non-negativity of $\gamma_i$ is all we need.

## G.5 DISCUSSIONS ON OTHER POLYNOMIAL BASES

The analysis in Section G.4 reflects the generality and flexibility of our proposed reparameterization technique, and we consider utilizing the simple reparameterization to conduct experiments on Bernstein and Monomial bases. The results are listed as POLYGCL (Bern) and POLYGCL (Mono) in Table 18. We conclude that the results of POLYGCL (Bern) and POLYGCL (Mono) are comparable with POLYGCL on certain datasets, which reflects the generality of our framework. In practice, we employ Chebyshev polynomials in POLYGCL due to comprehensive considerations of efficiency and effectiveness. This choice is supported by Table 19, which presents the average running time per epoch (ms) of POLYGCL using different bases.

## G.6 OTHER CORRUPTION METHODS

In our implementation of generating negative embeddings in POLYGCL, we simply shuffle the node features randomly. Further, we conduct experiments on real-world datasets to explore the impact

Table 18: Mean node classification accuracy (%) on real-world graphs as for different bases.

| Methods | Cora | Citeseer | Pubmed | Cornell | Texas | Wisconsin | Actor | Chameleon | Squirrel |
|---|---|---|---|---|---|---|---|---|---|
| POLYGCL | **87.57** $_{\pm0.62}$ | **79.81** $_{\pm0.85}$ | **87.15** $_{\pm0.27}$ | 82.62 $_{\pm3.11}$ | **88.03** $_{\pm1.80}$ | **85.50** $_{\pm1.88}$ | **41.15** $_{\pm0.88}$ | **71.62** $_{\pm0.96}$ | **56.49** $_{\pm0.72}$ |
| POLYGCL (Bern) | 85.51 $_{\pm0.69}$ | 77.48 $_{\pm0.70}$ | 83.90 $_{\pm0.24}$ | **83.13** $_{\pm3.01}$ | 80.23 $_{\pm2.14}$ | 82.40 $_{\pm2.75}$ | 38.08 $_{\pm0.88}$ | 69.35 $_{\pm1.12}$ | 51.76 $_{\pm0.94}$ |
| POLYGCL (Mono) | 84.94 $_{\pm1.01}$ | 78.52 $_{\pm0.78}$ | 85.49 $_{\pm0.33}$ | 79.87 $_{\pm2.18}$ | 83.27 $_{\pm3.05}$ | 81.42 $_{\pm2.50}$ | 39.19 $_{\pm1.01}$ | 66.41 $_{\pm1.20}$ | 49.45 $_{\pm0.79}$ |

Table 19: Average running time per epoch (ms) as for different bases.

| Methods | Cora | Citeseer | Pubmed | Cornell | Texas | Wisconsin | Actor | Chameleon | Squirrel |
|---|---|---|---|---|---|---|---|---|---|
| POLYGCL | 82.32 | 136.21 | 242.92 | 55.23 | 45.91 | 49.77 | 166.17 | 221.73 | 824.76 |
| POLYGCL (Bern) | 208.53 | 322.73 | 815.07 | 97.61 | 90.83 | 104.38 | 383.9 | 589.41 | 2284.05 |
| POLYGCL (Mono) | 105.44 | 167.95 | 180.48 | 46.71 | 38.63 | 45.04 | 130.62 | 245.82 | 596.69 |

of other data augmentation methods in POLYGCL. We denote edge-dropping as "ED", feature-masking as "FM", and subgraph-sampling as "SS" for short. Note that we follow the settings in CCA-SSG (Zhang et al., 2021) and GRACE (Zhu et al., 2020b) to perform edge-dropping and feature-masking at the same time, which is denoted as "ED&FM", and we perform the subgraph-sampling (SS) perturbation following GraphCL (You et al., 2020).

Table 20: Mean node classification accuracy (%) on real-world graphs as for different corruption methods.

| Methods | Cora | Citeseer | Pubmed | Cornell | Texas | Wisconsin | Actor | Chameleon | Squirrel |
|---|---|---|---|---|---|---|---|---|---|
| POLYGCL | **87.57** $_{\pm0.62}$ | **79.81** $_{\pm0.85}$ | **87.15** $_{\pm0.27}$ | 82.62 $_{\pm3.11}$ | **88.03** $_{\pm1.80}$ | **85.50** $_{\pm1.88}$ | **41.15** $_{\pm0.88}$ | **71.62** $_{\pm0.96}$ | **56.49** $_{\pm0.72}$ |
| POLYGCL (ED&FM) | 86.85 $_{\pm0.77}$ | 78.23 $_{\pm0.54}$ | 85.85 $_{\pm0.26}$ | **84.11** $_{\pm2.97}$ | 85.80 $_{\pm1.85}$ | 81.26 $_{\pm2.25}$ | 38.44 $_{\pm0.90}$ | 70.30 $_{\pm1.04}$ | 53.88 $_{\pm1.22}$ |
| POLYGCL (SS) | 84.74 $_{\pm0.84}$ | 75.30 $_{\pm0.79}$ | 82.61 $_{\pm0.28}$ | 80.33 $_{\pm1.80}$ | 82.62 $_{\pm3.11}$ | 76.25 $_{\pm2.25}$ | 33.10 $_{\pm1.26}$ | 65.84 $_{\pm1.42}$ | 46.04 $_{\pm0.81}$ |

In Table 20, we observe that the ED&FM perturbation slightly deteriorates the performance of POLYGCL but still seems comparable. However, the SS perturbation causes significant damage to the effectiveness of POLYGCL. We attribute this phenomenon to excessive perturbations in the graph topology and the loss of important spectrum information while conducting the subgraph-sampling operation.

## G.7 THE IMPORTANCE OF DECOUPLING MECHANISM

To demonstrate the effectiveness of the decoupling mechanism in POLYGCL, we report the mean accuracy results of the cSBM datasets in Table 21 as a comparison between POLYGCL and POLYGCL (Cheb). Note that in POLYGCL (Cheb), we directly apply GCL to the Chebyshev polynomial filters without decoupling the low-pass and high-pass information. POLYGCL generally outperforms POLYGCL (Cheb) on different homophily levels, especially in extreme homophilic/heterophilic settings ($|\phi| \to 1$).

Further, we visualize the normalized learned filters of POLYGCL (Cheb) on the cSBM datasets in Figure 5. Compared with Figure 4, POLYGCL (Cheb) learns complex filters with a high degree of oscillation across homophily, while the filter curve learned by POLYGCL is smoother. From the spectral perspective, this observation supports our preference for the training paradigm in POLYGCL over learning spectral filters directly via GCL.

Table 21: Mean node classification accuracy (%) with a 95% confidence interval on cSBM graphs: Decoupling Analysis.

| Methods | $\phi = -1$ | $\phi = -0.75$ | $\phi = -0.5$ | $\phi = -0.25$ | $\phi = 0$ | $\phi = 0.25$ | $\phi = 0.5$ | $\phi = 0.75$ | $\phi = 1$ |
|---|---|---|---|---|---|---|---|---|---|
| POLYGCL | $98.84_{\pm0.17}$ | $94.23_{\pm0.31}$ | $90.82_{\pm0.50}$ | $75.43_{\pm0.68}$ | $66.51_{\pm0.69}$ | $69.43_{\pm0.65}$ | $88.22_{\pm0.72}$ | $98.09_{\pm0.29}$ | $99.29_{\pm0.23}$ |
| POLYGCL (Cheb) | $79.28_{\pm0.46}$ | $85.35_{\pm0.82}$ | $81.04_{\pm0.57}$ | $77.27_{\pm0.90}$ | $65.90_{\pm0.53}$ | $70.34_{\pm0.98}$ | $84.63_{\pm0.81}$ | $92.97_{\pm0.33}$ | $89.72_{\pm0.58}$ |

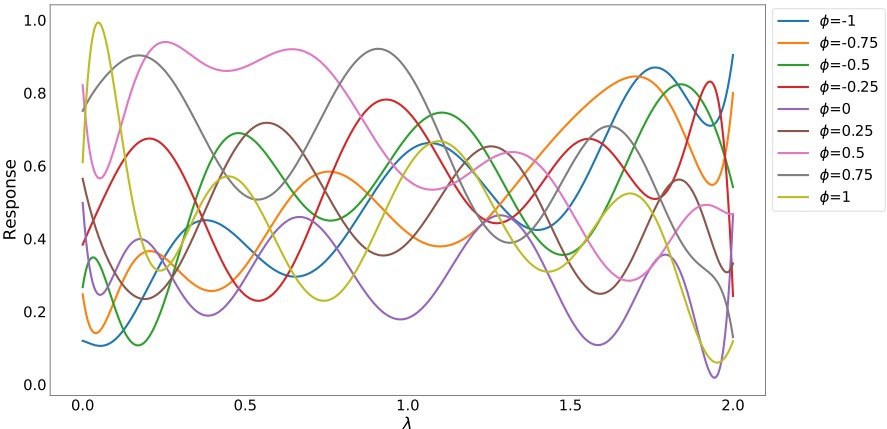

Figure 5: The normalized learned filters of POLYGCL (Cheb) on cSBM datasets.

