# OpenReview forum: "PolyGCL: GRAPH CONTRASTIVE LEARNING via Learnable Spectral Polynomial Filters"
_ICLR.cc/2024/Conference — ICLR 2024 spotlight_

### Official Review · Reviewer_LQgR · 2023-10-17

**Soundness:** 3 good
**Presentation:** 3 good
**Contribution:** 2 fair
**Rating:** 8
**Confidence:** 4

**Summary:**

This paper focuses on self-supervised learning applied to spectral GNNs. It addresses the limitation of existing contrastive learning techniques when dealing with high-pass heterophilic graphs. The authors tackle this challenge by introducing high-pass information in graph contrastive learning (GCL) and propose PolyGCL, a GCL pipeline that leverages polynomial graph basis to generate different spectral views for contrastive learning. Experimental results on synthetic and realworld datasets validate the effectiveness of the proposed PolyGCL.

**Strengths:**

+ The introduction of polynomial spectral GNNs into graph contrastive learning is a novel and interesting idea.
+ PolyGCL is well-motivated, and the authors provide empirical evidence to support the necessity of designing specific polynomial filter structures.
+ There are some decent theoretical analyses of the proposed method.
+ Experimental results demonstrate the effectiveness of PolyGCL.

**Weaknesses:**

- Certain part of the introduced model need further clarrification. The authors propose to optimize a high-pass and a low-pass filter with polynomial spectral filters respectively, but ultimately arrive at a linear combination of the outputs of these two filter results. However, the optimization target in Equation (4) doesn't involve $\alpha$ and $\beta$ at all .
- The rationale behind randomly shuffling the node feature to obtain the contrastive view needs to be explained, since is not a common practice to pertub graph representation.
- In section 4.4, authors conclude that the aggregation of low-pass and high-pass representations achieves better error upper bound compared with just one aspect of representation. However, it still doesn't explain PolyGCL's advantage over directaly applying GCL on normal spectral GNNs like ChebNet.
- The paper lacks sufficient ablation studies on the two separate filters, such as how the model's performance is affected when there is only one filter left or when $\alpha$ and $\beta$ are fixed.

**Questions:**

1. Please provide a more detailed explanation of how $\alpha$ and $\beta$ are optimized in Equation (4). Are there any supervising signals to optimize these two parameters? If so, the model seems to be equivalant to directaly optimizing a polynomial filter since all of the coefficients are learnable.
2. Please give the rationale for randomly shuffling node features as a pertubation method. Would the model performance be influenced if the random shuffle were replaced with other methods like edge-dropping or subgraph-sampling?
3. Please provide more explanations on PolyGCL's advantage over directaly applying GCL on spectral GNNs?
4. How would the model performance when there is only one filter left or $\alpha$ and $\beta$ are fixed?

---

> ### Author Response · Authors · 2023-11-17
> **Reply to Reviewer LQgR (1)**
>
> Thank you for the detailed comments and valuable questions. We provide details to clarify your major concerns.
>
> The reply to weakness is integrated into the reply to questions as follows.
>
> **Reply to Questions**
>
>
> >Q1: Please provide a more detailed explanation of how $\alpha$ and $\beta$ are optimized in Equation (4). Are there any supervising signals to optimize these two parameters? If so, the model seems to be equivalant to directaly optimizing a polynomial filter since all of the coefficients are learnable.
>
> A1: To make it clearer, we consider rewriting Equation (4) in Section 4.2 as follows to explicitly reflect the optimization process of parameters $\alpha, \beta$.
> $$
> \begin{aligned}
> \mathcal{L}_{\text{BCE}}  &=\frac{1}{4 N}\left(\sum\_{i=1}^N \log \mathcal{D}\left(\mathbf{Z}_L^i, \mathbf{g}\right)+\log \left(1-\mathcal{D}\left(\tilde{\mathbf{Z}}_L^i, \mathbf{g}\right)\right)+ \log \mathcal{D}\left(\mathbf{Z}_H^i, \mathbf{g}\right)+\log \left(1-\mathcal{D}\left(\tilde{\mathbf{Z}}_H^i, \mathbf{g}\right)\right)\right)
> \\\\
> &=\frac{1}{4 N}\left(\sum\_{i=1}^N \log \mathcal{D}\left(\mathbf{Z}_L^i, \textbf{Mean}(\mathbf{Z})\right)+\log \left(1-\mathcal{D}\left(\tilde{\mathbf{Z}}_L^i, \textbf{Mean}(\mathbf{Z})\right)\right)\right. \\\\
> & \quad \quad \quad \quad + \left.\log \mathcal{D}\left(\mathbf{Z}_H^i, \textbf{Mean}(\mathbf{Z})\right)+\log \left(1-\mathcal{D}\left(\tilde{\mathbf{Z}}_H^i, \textbf{Mean}(\mathbf{Z})\right)\right)\right)\\\\
> &=\frac{1}{4 N}\left(\sum\_{i=1}^N \log \mathcal{D}\left(\mathbf{Z}_L^i, \textbf{Mean}(\alpha \mathbf{Z}_L+\beta \mathbf{Z}_H)\right)+\log \left(1-\mathcal{D}\left(\tilde{\mathbf{Z}}_L^i, \textbf{Mean}(\alpha \mathbf{Z}_L+\beta \mathbf{Z}_H)\right)\right)\right. \\\\
> & \quad \quad \quad \quad + \left.\log \mathcal{D}\left(\mathbf{Z}_H^i, \textbf{Mean}(\alpha \mathbf{Z}_L+\beta \mathbf{Z}_H)\right)+\log \left(1-\mathcal{D}\left(\tilde{\mathbf{Z}}_H^i, \textbf{Mean}(\alpha \mathbf{Z}_L+\beta \mathbf{Z}_H)\right)\right)\right).
> \end{aligned}
> $$
>
> As shown in the above equation, the optimization target involves the learnable parameters $\alpha$  and $\beta$. Further, considering that all of the coefficients are learnable, PolyGCL can be viewed as directly optimizing a polynomial filter. We claim that compared with directly optimizing a polynomial filter in self-supervised settings, the training paradigm in PolyGCL makes the polynomial filter easier to fit the filter functions and exhibits better adaption to graphs across homophily due to the decoupling of low-pass and high-pass information. Please refer to Q3 for detailed explanations.

---

> ### Author Response · Authors · 2023-11-17
> **Reply to Reviewer LQgR (2)**
>
> >Q2: Please give the rationale for randomly shuffling node features as a pertubation method. Would the model performance be influenced if the random shuffle were replaced with other methods like edge-dropping or subgraph-sampling?
>
> A2: We explain the rationale for randomly shuffling node features as a perturbation method as follows.
>
> - Firstly, randomly shuffling node features can be considered as an easy-to-implement way to construct corrupted samples, and a series of works (DGI [1], MVGRL [2], GraphCL [3], GGD [4]) adopt this simple strategy and demonstrate effectiveness in practice.
> - Besides, randomly shuffling features is an operation at the node feature level without destroying the structural information, thus will not cause perturbation to the graph spectrum. This property facilitates our analysis of SRL in Section 4.3.
>
> Further, we conduct experiments on real-world datasets to explore the impact of other data augmentation methods on PolyGCL. We denote edge-dropping as "ED", feature-masking as "FM", and subgraph-sampling as "SS" for short. Note that we follow the settings in CCA-SSG [5] and GRACE [6] to perform edge-dropping and feature-masking at the same time, which is denoted as "ED\&FM", and we perform the subgraph-sampling (SS) perturbation following GraphCL [3].
>
> | Methods                | cora                | citeseer            | pubmed              | cornell             | texas               | wisconsin           | actor               | chameleon           | squirrel            |
> |------------------------|---------------------|---------------------|---------------------|---------------------|---------------------|---------------------|---------------------|---------------------|---------------------|
> | PolyGCL               | 87.57 ± 0.62        | 79.81 ± 0.85    | 87.15 ± 0.27    | 82.62 ± 3.11        | 88.03 ± 1.80    | 85.50 ± 1.88    | 41.15 ± 0.88    | 71.62 ± 0.96    | 56.49 ± 0.72    |
> | PolyGCL(ED&FM)         | 86.85 ± 0.77        | 78.23 ± 0.54        | 85.85 ± 0.26        | 84.11 ± 2.97    | 85.80 ± 1.85        | 81.26 ± 2.25        | 38.44 ± 0.90    | 70.30 ± 1.04    | 53.88 ± 1.22    |
> | PolyGCL(SS)            | 84.74 ± 0.84        | 75.30 ± 0.79        | 82.61 ± 0.28        | 80.33 ± 1.80        | 82.62 ± 3.11        | 76.25 ± 2.25        | 33.10 ± 1.26        | 65.84 ± 1.42        | 46.04 ± 0.81        |
>
>
> In the above table, we observe that the ED\&FM perturbation slightly deteriorates the performance of PolyGCL but still seems comparable. However, the SS perturbation causes significant damage to the effectiveness of PolyGCL. We attribute this phenomenon to excessive perturbations in the graph topology and the loss of important spectrum information while conducting the subgraph-sampling operation.

---

> ### Author Response · Authors · 2023-11-17
> **Reply to Reviewer LQgR (3)**
>
> >Q3: Please provide more explanations on PolyGCL's advantage over directaly applying GCL on spectral GNNs?
>
>
> A3: In Section 4.4, we provide a theoretical analysis of decoupling the low-pass and high-pass representations and linearly combining them, which guarantees a better error upper bound compared with just one aspect of representation.  It is noted that all the discussions in Section 4.4 are based on the decoupling of the low-pass and high-pass information, which conforms to the learning process of PolyGCL. Previous works have pointed out that spectral polynomial filters have the capability of learning filters of any shape, however, their learning paradigm highly relies on the annotated data, which corresponds to supervised learning. As shown in Figure 1, in the self-supervised setting, directly applying GCL on spectral GNNs results in performance degradation.
>
> | Methods               | $\phi=-1$            | $\phi=-0.75$         | $\phi=-0.5$         | $\phi=-0.25$         | $\phi=0$            | $\phi=0.25$         | $\phi=0.5$           | $\phi=0.75$          | $\phi=1$             |
> |-----------------------|-----------------|-----------------|-----------------|-----------------|-----------------|-----------------|-----------------|-----------------|-----------------|
> | PolyGCL              | 98.84 ± 0.17    | 94.23 ± 0.31    | 90.82 ± 0.50    | 75.43 ± 0.68    | 66.51 ± 0.69    | 69.43 ± 0.65    | 88.22 ± 0.72    | 98.09 ± 0.29    | 99.29 ± 0.23    |
> | PolyGCL(Cheb)         | 79.28 ± 0.46    | 85.35 ± 0.82    | 81.04 ± 0.57    | 77.27 ± 0.90    | 65.90 ± 0.53    | 70.34 ± 0.98    | 84.63 ± 0.81    | 92.97 ± 0.33    | 89.72 ± 0.58    |
>
>
> We also report the mean accuracy results of the cSBM datasets in the above table as a comparison between PolyGCL and PolyGCL(Cheb), which directly applies GCL to the Chebshev polynomial filters. PolyGCL generally outperforms PolyGCL(Cheb) on different homophily levels, especially in extreme homophilic/heterophilic settings ($|\phi|\to 1$).
>
> Further, we visualize the learned filters in PolyGCL(Cheb) on the cSBM datasets. Compared Figure 5 in Appendix E.4 with Figure 4 in Section 5.4, we conclude that although PolyGCL(Cheb) can roughly capture the low-pass and high-pass tendency for homophilic and heterophilic graphs respectively, it tends to learn the complex filters with a high degree of oscillation across homophily, which exhibits difficulty with the absence of supervised signals.
>
> As a comparison, the filter curve learned in Figure 4 is smoother, and its low-pass/high-pass trend is more significant under the homophilic/heterophilic settings, which shows the effectiveness of the learning paradigm in PolyGCL, that is, decoupling the low-pass and high-pass information and linearly combining them later to obtain the final embeddings.

---

> ### Author Response · Authors · 2023-11-17
> **Reply to Reviewer LQgR (4)**
>
> >Q4: How would the model performance when there is only one filter left or $\alpha$ and $\beta$ are fixed?
>
> A4: We include experiments where $\alpha$ and $\beta$ are set to zero separately as the ablation studies. Specifically, $\alpha=0, \beta=0$ means only the high-pass/low-pass filter is reserved respectively. The results are given in the following table.
>
> | Methods                | cora                | citeseer            | pubmed              | cornell             | texas               | wisconsin           | actor               | chameleon           | squirrel            |
> |------------------------|---------------------|---------------------|---------------------|---------------------|---------------------|---------------------|---------------------|---------------------|---------------------|
> | PolyGCL               | 87.57 ± 0.62        | 79.81 ± 0.85    | 87.15 ± 0.27    | 82.62 ± 3.11        | 88.03 ± 1.80    | 85.50 ± 1.88    | 41.15 ± 0.88    | 71.62 ± 0.96    | 56.49 ± 0.72    |
> | PolyGCL($\alpha=0$)           | 70.67 ± 0.48        | 64.22 ± 0.93        | 76.41 ± 0.35        | 80.16 ± 2.62        | 86.52 ± 1.97        | 82.07 ± 2.75        | 38.28 ± 0.39        | 68.21 ± 1.40        | 52.10 ± 0.80        |
> | PolyGCL($\beta=0$)           | 87.65 ± 0.67    | 78.76 ± 0.75        | 86.64 ± 0.17        | 76.23 ± 5.41        | 82.56 ± 2.13        | 68.88 ± 2.50        | 37.36 ± 0.46        | 65.08 ± 1.27        | 48.52 ± 0.71        |
>
> Note that PolyGCL achieves better performance when optimized over both the parameters in most datasets. In addition, we observe that PolyGCL($\alpha=0$) remains comparable in heterophilic datasets, while PolyGCL($\beta=0$) shows better adaptability to homophilic settings. The results are further consistent with the low-pass/high-pass preference for homophilic/heterophilic settings in Figure 3 in Section 5.4.
>
> References:
>
> [1] Veličković, Petar, et al. "Deep graph infomax." _arXiv preprint arXiv:1809.10341_ (2018).
>
> [2] Hassani, Kaveh, and Amir Hosein Khasahmadi. "Contrastive multi-view representation learning on graphs." _International conference on machine learning_. PMLR, 2020.
>
> [3] You, Yuning, et al. "Graph contrastive learning with augmentations." _Advances in neural information processing systems_ 33 (2020): 5812-5823.
>
> [4] Zheng, Yizhen, et al. "Rethinking and scaling up graph contrastive learning: An extremely efficient approach with group discrimination." _Advances in Neural Information Processing Systems_ 35 (2022): 10809-10820.
>
> [5] Zhang, Hengrui, et al. "From canonical correlation analysis to self-supervised graph neural networks." _Advances in Neural Information Processing Systems_ 34 (2021): 76-89.
>
> [6] Zhu, Yanqiao, et al. "Deep graph contrastive representation learning." _arXiv preprint arXiv:2006.04131_ (2020).

---

> > ### Comment · Reviewer_LQgR · 2023-11-20
> >
> > I appreciate the authors efforts in clarifying the details as well as providing extra expariments. Since all my concerns have been addressed, I will update the score and support this work.

---

> > > ### Author Response · Authors · 2023-11-20
> > > **Official Comment by Authors**
> > >
> > > We sincerely thank the reviewer for the time and effort invested in re-evaluating our work. We will also include more results following the suggestions in the revised version.

---

### Official Review · Reviewer_2heL · 2023-10-30

**Soundness:** 4 excellent
**Presentation:** 2 fair
**Contribution:** 4 excellent
**Rating:** 8
**Confidence:** 3

**Summary:**

The paper introduces a novel approach to Graph Contrastive Learning (GCL) named POLYGCL, designed to address the challenges of applying GCL to both homophilic and heterophilic graphs. The primary limitation addressed is the inherent smoothness introduced by traditional low-pass GNN encoders. POLYGCL employs learnable spectral polynomial filters to balance between low-pass and high-pass views, providing better representation learning for both graph types. Theoretical proofs underscore the effectiveness of this combined filter approach, and empirical evaluations on synthetic and real-world datasets validate the method's superiority over existing GCL paradigms. The research fills a crucial gap in the realm of graph representation learning, particularly concerning heterophilic graphs.

**Strengths:**

1. **Originality**:
    - The work presents a novel approach in the form of **PolyGCL** that combines both low-pass and high-pass filters in graph-based learning.
    - The theoretical foundations are robust, and the introduction of a combined filter in graph learning is a commendable initiative.
    - The work extends beyond existing literature by addressing both Graph Contrastive Learning (GCL) methods and spectral-based GNNs, which is an original contribution to the domain.

2. **Quality**:
    - The paper is backed by rigorous theoretical formulations, evidenced by multiple theorems, remarks, propositions, and corollaries.
    - The empirical validations, including both synthetic and real-world datasets, provide strong support for the paper's claims. A diverse set of baselines have been chosen for comparisons, ensuring a comprehensive evaluation.

3. **Significance**:
    - The paper addresses challenges in existing GCL methods and spectral-based GNNs, a pressing need in the domain of graph representation learning.
    - The proposed method's prowess on heterophilic graphs is especially significant given the inherent challenges of these graph types.
    - The complexity analysis suggests the scalability of the method, highlighting its potential applicability to large-scale problems in the future.

---

In conclusion, the paper stands out in its originality, depth of analysis, and significance to the graph learning community. It is a valuable contribution to the literature and holds promise for further explorations in this direction.

**Weaknesses:**

1. **Lack of Intuitive Explanation for Choice of Chebyshev Polynomials:** The paper introduces the use of Chebyshev polynomials without providing a deep intuitive rationale behind this choice. For readers unfamiliar with Chebyshev polynomials or their relevance in graph-based learning, this can be a point of confusion. A discussion on why Chebyshev polynomials were chosen over other potential polynomial bases could add depth to the work.

2. **Decoupling of Low-Pass and High-Pass Information**: While the paper emphasizes the importance of decoupling low-pass and high-pass filter functions, it doesn't thoroughly discuss the potential pitfalls or challenges of this approach. It would be beneficial to discuss under which conditions this decoupling might not be advantageous or how it compares to methods that don't employ this decoupling.

3. **Reparameterization and Assumptions**: The paper introduces reparameterization techniques, such as using the prefix sum and prefix difference, to ensure non-negativity and a decremental filter value. However, there's a lack of clarity on the implications of these assumptions and reparameterizations on the model's performance and generality. It would be beneficial to delve deeper into why these particular reparameterization techniques were chosen and their impact on the robustness and flexibility of the approach.

**Questions:**

1. **Motivation behind Chebyshev Polynomials**: Can the authors provide more insight into why Chebyshev polynomials were specifically chosen as the base polynomials? How do they compare to other potential polynomial bases in terms of efficacy and computational efficiency in the context of Graph Contrastive Learning?

2. **Filter Decoupling Challenges**: Are there potential challenges or scenarios where the decoupling of low-pass and high-pass information might be less advantageous? How does the method account for such scenarios?

3. **Limitations of the Model**: Every model has its limitations. What would the authors identify as the key limitations of the PolyGCL method? Are there specific types of graph structures or data distributions where PolyGCL might not be the best choice?

4. **Future Work**: Given the current findings and the proposed method, what do the authors see as the next steps or future directions in this line of research? Are there plans to extend PolyGCL or integrate it with other techniques to address more complex graph learning scenarios?

---

> ### Author Response · Authors · 2023-11-17
> **Reply to Reviewer 2heL (1)**
>
> Thanks for your attention and interest in our work, and we are grateful for your valuable and constructive feedback.
>
> **Reply to Weakness**
>
>
> W1 and W2 are integrated into Q1 and Q2 respectively.
>
> >W3: Reparameterization and Assumptions: The paper introduces reparameterization techniques, such as using the prefix sum and prefix difference, to ensure non-negativity and a decremental filter value. However, there's a lack of clarity on the implications of these assumptions and reparameterizations on the model's performance and generality. It would be beneficial to delve deeper into why these particular reparameterization techniques were chosen and their impact on the robustness and flexibility of the approach.
>
> A1:  We utilize the reparameterization techniques to ensure the low-pass and high-pass properties of the learned filters during the learning process, which correspond to the filter functions with incremental and decremental values, respectively.
>
> Firstly, in spectral analysis, low-pass/high-pass filters are better defined when the value of filter function is non-negative, and the advantages of ensuing non-negativity are also discussed in BernNet [1]. Based on the non-negativity, the prefix sum/difference is introduced to ensure the low-pass/high-pass property of the filter functions learned by polynomials during model training, which makes the low-pass and high-pass information decouple naturally and further promotes the effective learning of the separate filters to better adapt to homophily/heterophily scenarios.
>
> We list the comparison between the results of PolyGCL and PolyGCL(wo-RP) in the following table, where PolyGCL(wo-RP) denotes PolyGCL **w**ith**o**ut **R**e**P**arameterization. By decoupling the low-pass and high-pass information via the simple reparameterization, PolyGCL benefits from more stable model training process and improves the performance on homophilic/heterophilic datasets.
>
> | Methods             | cora              | citeseer          | pubmed            | cornell           | texas             | wisconsin         | actor             | chameleon         | squirrel          |
> |---------------------|-------------------|-------------------|-------------------|-------------------|-------------------|-------------------|-------------------|-------------------|-------------------|
> | PolyGCL           | 87.57 ± 0.62      | 79.81 ± 0.85  | 87.15 ± 0.27  | 82.62 ± 3.11      | 88.03 ± 1.80  | 85.50 ± 1.88  | 41.15 ± 0.88  | 71.62 ± 0.96  | 56.49 ± 0.72  |
> | PolyGCL(wo-RP)      | 85.09 ± 0.78      | 76.74 ± 0.80      | 84.39 ± 0.29      | 78.26 ± 3.02      | 84.11 ± 2.29      | 79.50 ± 3.50      | 38.14 ± 0.96      | 65.98 ± 0.95      | 50.06 ± 1.01      |
>
>
> In addition, the reparameterization technique based on non-negativity and prefix sum/difference can be easily extended to other polynomial bases, which further verifies the robustness of this technique. In detail, as for the Bernstein polynomial, which can also learn arbitrary filters, we directly utilize this reparameterization on the coefficients $\theta_k$ which proves to be equivalent to the filter value $h(\lambda)$. Besides, as for the Monomial polynomial in GPR-GNN [2], we can also consider non-negative coefficients to ensure the low-pass/high-pass property, for instance, $\sum_{i=0}^K \gamma_i (2\mathbf{I}-\mathbf{L})^i$ and $\sum_{i=0}^K \gamma_i \mathbf{L}^i$ for low-pass/high-pass filters respectively, where the non-negativity of $\gamma_i$ is all we need.
>
> The above analysis reflects the generality and flexibility of our proposed reparameterization technique, and we consider utilizing the simple reparameterization to conduct experiments on Bernstein and Monomial bases, the results are listed as PolyGCL(Bern) and PolyGCL(Mono) in Q1.

---

> ### Author Response · Authors · 2023-11-17
> **Reply to Reviewer 2heL (2)**
>
> **Reply to Questions**
>
>
> > Q1: Motivation behind Chebyshev Polynomials: Can the authors provide more insight into why Chebyshev polynomials were specifically chosen as the base polynomials? How do they compare to other potential polynomial bases in terms of efficacy and computational efficiency in the context of Graph Contrastive Learning?
>
> A2: We summarize the reasons for choosing Chebyshev polynomials as the base polynomials as follows.
>
> - Theoretically, ChebyNetII [3] and [4] claims that the Chebyshev polynomial achieves the optimum convergent rate and near-optimum error when approximating a function compared with other bases.
> -  Both Chebshev and Monomial bases are linear time complexity related to propagation step $K$, but BernNet is quadratic time complexity related to $K$.
> - In practice, we utilize the Chebyshev polynomials out of comprehensive considerations of efficiency and effectiveness, as shown in the following tables which report the mean accuracy and the average run time per epoch (ms) of the models.
>
> | Methods             | cora              | citeseer          | pubmed            | cornell           | texas             | wisconsin         | actor             | chameleon         | squirrel          |
> |---------------------|-------------------|-------------------|-------------------|-------------------|-------------------|-------------------|-------------------|-------------------|-------------------|
> | PolyGCL            | 87.57 ± 0.62  | 79.81 ± 0.85  | 87.15 ± 0.27  | 82.62 ± 3.11      | 88.03 ± 1.80  | 85.50 ± 1.88  | 41.15 ± 0.88  | 71.62 ± 0.96  | 56.49 ± 0.72  |
> | PolyGCL(Bern)       | 85.51 ± 0.69      | 77.48 ± 0.70      | 83.90 ± 0.24      | 83.13 ± 3.01  | 80.23 ± 2.14      | 82.40 ± 2.75  | 38.08 ± 0.88      | 69.35 ± 1.12  | 51.76 ± 0.94  |
> | PolyGCL(Mono)       | 84.94 ± 1.01      | 78.52 ± 0.78  | 85.49 ± 0.33      | 79.87 ± 2.18      | 83.27 ± 3.05      | 81.42 ± 2.50      | 39.19 ± 1.01      | 66.41 ± 1.20      | 49.45 ± 0.79      |
>
>
> | Methods             | cora    | citeseer | pubmed  | cornell | texas  | wisconsin | actor   | chameleon | squirrel |
> |---------------------|---------|----------|---------|---------|--------|-----------|---------|-----------|----------|
> | PolyGCL            | 82.32   | 136.21   | 242.92  | 55.23   | 45.91  | 49.77     | 166.17  | 221.73    | 824.76   |
> | PolyGCL(Bern)       | 208.53  | 322.73   | 815.07  | 97.61   | 90.83  | 104.38    | 383.9   | 589.41    | 2284.05  |
> | PolyGCL(Mono)       | 105.44  | 167.95   | 180.48  | 46.71   | 38.63  | 45.04     | 130.62  | 245.82    | 596.69   |
>
>
> Besides, based on the above tables, we conclude that the results of PolyGCL(Bern) and PolyGCL(Mono) are comparable with PolyGCL on certain datasets, which reflects the generality of our framework.
>
>
> > Q2: Filter Decoupling Challenges: Are there potential challenges or scenarios where the decoupling of low-pass and high-pass information might be less advantageous? How does the method account for such scenarios?
>
>
> A3:  Yes. For exmaple, if the graph is extremely homophilic, there is almost no need for high-pass frequency. In this case, the introduction of high-pass information tends to be less advantageous. However, PolyGCL has the potential to adjust the relative value between the linear coefficients $\alpha$ and $\beta$ adaptively.
>
> As shown in Figure 3 in Section 5.4, when $\phi \to 1$ (which indicates the extreme homophilic settings in cSBM), the linear coefficient $\alpha$ for the low-pass filter approaches $1$, meaning the value of $\beta$ for the high-pass filter approaches $0$ and PolyGCL degenerates into considering only low-pass information. However, most real graphs will not be such an extreme case, thus PolyGCL performs well on real datasets.

---

> ### Author Response · Authors · 2023-11-17
> **Reply to Reviewer 2heL (3)**
>
> > Q3: Limitations of the Model: Every model has its limitations. What would the authors identify as the key limitations of the PolyGCL method? Are there specific types of graph structures or data distributions where PolyGCL might not be the best choice?
>
> A4: The key limitation of PolyGCL lies in its scalability to large graphs. In Section 5.5, the time complexity of PolyGCL is linear to $N$ and $E$, which represent the number of nodes and edges respectively. Compared with other baselines in Table 4 in Section 5.5, PolyGCL can handle graphs with a medium size ($10^5-10^6$ nodes) due to its linear complexity and the pre-computing of polynomial filters. However, PolyGCL still suffers from OOM when modeling extremely large graphs with over 100 million nodes (ogbn-papers100M [5]).
>
> Besides, there are a series of works [6, 7] showing that the homophily pattern of graphs is actually very complex and may be local or not critical. Spectrum analysis in these cases is still lacking, and the low-pass/high-pass filtering may not be the decisive factor in the applicability of homophily/heterophily, which limits the expressiveness of PolyGCL.
>
> In addition, as for the graph with too many noise edges that destroy its spectral, PolyGCL which relies on the spectral filters might not be the best choice. In this case, the node features themselves are of more significance to some extent.
>
>
> >Q4: Future Work: Given the current findings and the proposed method, what do the authors see as the next steps or future directions in this line of research? Are there plans to extend PolyGCL or integrate it with other techniques to address more complex graph learning scenarios?
>
> A5: We believe that performing efficient self-supervised learning on extremely large graphs (ogbn-papers100M [5]) with spectral analysis is a promising future direction in this line of research. Considering the linear complexity and the nature of pre-computing polynomial propagation, PolyGCL exhibits the potential to further extend its scalability.
>
> As for more complex graph learning scenarios, we intend to explore the complex homophily pattern of graphs shown in [6, 7] and the spectral analysis of heterogeneous graphs in self-supervised settings. We believe the spectral analysis in PolyGCL will play a key role or open up a new window for these complex scenarios.
>
> References:
>
> [1] He, Mingguo, Zhewei Wei, and Hongteng Xu. "Bernnet: Learning arbitrary graph spectral filters via bernstein approximation." _Advances in Neural Information Processing Systems_ 34 (2021): 14239-14251.
>
> [2] Chien, Eli, et al. "Adaptive universal generalized pagerank graph neural network." _arXiv preprint arXiv:2006.07988_ (2020).
>
> [3] He, Mingguo, Zhewei Wei, and Ji-Rong Wen. "Convolutional neural networks on graphs with chebyshev approximation, revisited." _Advances in Neural Information Processing Systems_ 35 (2022): 7264-7276.
>
> [4] Geddes, Keith O. "Near-minimax polynomial approximation in an elliptical region." _SIAM Journal on Numerical Analysis_ 15.6 (1978): 1225-1233.
>
> [5] Weihua Hu, Matthias Fey, Marinka Zitnik, Yuxiao Dong, Hongyu Ren, Bowen Liu, Michele Catasta, and Jure Leskovec. Open graph benchmark: Datasets for machine learning on graphs. Advances in neural information processing systems, 33:22118–22133, 2020.
>
> [6] Mao, Haitao, et al. "Demystifying Structural Disparity in Graph Neural Networks: Can One Size Fit All?." _arXiv preprint arXiv:2306.01323_ (2023).
>
> [7] Luan, Sitao, et al. "When do graph neural networks help with node classification: Investigating the homophily principle on node distinguishability." _arXiv preprint arXiv:2304.14274_ (2023).

---

> ### Author Response · Authors · 2023-11-23
> **A Kind Reminder: the author-reviewer discussion period is coming to an end.**
>
> Dear reviewer 2heL,
>
> We sincerely appreciate your thorough and detailed reviews of our submission. We hope that you will find our responses satisfactory and that they help clarify your concerns. We appreciate the opportunity to engage with you. We would like to kindly remind you that the discussion period is coming to an end. Could you please inform us whether our responses have addressed your concerns or if there are any other questions you need us to clarify?
>
> Thank you very much for your time.
>
> Best regards,
>
> Authors of Submission 7529

---

### Official Review · Reviewer_aFan · 2023-10-31

**Soundness:** 3 good
**Presentation:** 3 good
**Contribution:** 3 good
**Rating:** 8
**Confidence:** 3

**Summary:**

This paper addresses graph contrastive learning with learnable spectral filters, allowing to tackle homophilic and Heterophilic graphs.

**Strengths:**

Graph contrastive learning is of great interest, including the challenging goal of addressing heterophilic graphs.
The proposed idea is interesting and the paper providing theoretical and relevant experimental results is well written.

**Weaknesses:**

The main issue is that this paper is not well positioned within the literature of graph contrastive learning (GCL), including several missing work on GCL with spectral filters and/or adaptive/learnable filters, such as
* Liu, Nian, Xiao Wang, Deyu Bo, Chuan Shi, and Jian Pei. "Revisiting graph contrastive learning from the perspective of graph spectrum." Advances in Neural Information Processing Systems 35 (2022): 2972-2983.
* Xie, Xuanting, Wenyu Chen, Zhao Kang, and Chong Peng. "Contrastive graph clustering with adaptive filter." Expert Systems with Applications 219 (2023): 119645.
* Zhang, Hengrui, Qitian Wu, Yu Wang, Shaofeng Zhang, Junchi Yan, and Philip S. Yu. "Localized Contrastive Learning on Graphs." arXiv preprint arXiv:2212.04604 (2022).
The latter also provide some results on heterophilic graphs. Another paper on GCL with homophilic and heterophilic graphs is
* Wang, Haonan, Jieyu Zhang, Qi Zhu, and Wei Huang. "Can Single-Pass Contrastive Learning Work for Both Homophilic and Heterophilic Graph?." arXiv preprint arXiv:2211.10890 (2022).
and some theoretical results in this paper can be related to works conducted by other researchers
* HaoChen, Jeff Z., Colin Wei, Adrien Gaidon, and Tengyu Ma. "Provable guarantees for self-supervised deep learning with spectral contrastive loss." Advances in Neural Information Processing Systems 34 (2021): 5000-5011.

There are some spelling and grammatical errors, such as “in an self-supervised manner”, “with an self-supervised one”, “Trainging Algorithm”, “We denote Y … be the”, “ further justification are discussed”, “claims that maximize”, “optimising … also maximize”, “proves that maximize”, “Shannon diverge”

**Questions:**

Please position this work and its contributions with respect to the aforementioned papers

**Details Of Ethics Concerns:**

-

---

> ### Author Response · Authors · 2023-11-17
> **Reply to Reviewer aFan**
>
> Thanks for your valuable and constructive feedback!
>
> **Reply to Weakness**
>
>
> >Q1: Please position this work and its contributions with respect to the aforementioned papers.
>
> A1: Specifically, we include two new baselines SPGCL [1] and HLCL[2], which are GCLs that address heterophily. The results are shown in Appendix E.2 (Table 12, Table 13 and Table 14), which further shows the superiority of PolyGCL.
>
> Besides, we will further discuss the difference between PolyGCL and the GCL methods using spectral augmentations [3, 4]. SpCo [3] studies the necessity of high-frequency information in GCL and learns the optimal augmentation from the spectral view. [4] proposes a set of well-motivated graph transformation operations derived via graph spectral analysis, which are spectral graph cropping and graph frequency components reordering. Compared with [3, 4], PolyGCL does not require specially designed or complex preprocessing steps for spectrum augmentations but achieves the contrast between low-pass and high-pass information by directly optimizing the corresponding decoupled filters.
>
> As for [5] which also considers the spectral filters in self-supervised settings, we claim that PolyGCL has the ability to learn filters of any shape while [5] only considers a fixed set of polynomial filters, that is $(\mathbf{I}-\frac{\mathbf{L}}{2})^k$ and $(\frac{\mathbf{L}}{2})^k$, which restricts its expressiveness.
>
> From the theoretical view, [6] proposes the spectral contrastive loss which builds connections between the contrastive loss and the spectral method. In addition, Local-GCL [7] devises a kernelized contrastive loss with linear complexity for GCL, which also shows effectiveness in heterophilic graphs. However, these works related to the graph spectrum mainly focused on the analysis of eigenvalues while PolyGCL cares more about the learning of filter functions to adapt to homophilic/heterophilic settings.
>
> Compared with the above works, to the best of our knowledge, PolyGCL are the first to achieve efficient learning of low-pass and high-pass filters via polynomial approximation in a self-supervised setting. We will further include these works and conduct a more in-depth analysis compared with PolyGCL in the revised version.
>
> >Q2: There are some spelling and grammatical errors.
>
> A2: Thanks for pointing these spelling and grammatical errors out. We will go through the writings carefully in the revised version.
>
> References:
>
> [1] Wang, Haonan, et al. "Can Single-Pass Contrastive Learning Work for Both Homophilic and Heterophilic Graph?." _arXiv preprint arXiv:2211.10890_ (2022).
>
> [2] Yang, Wenhan, and Baharan Mirzasoleiman. "Contrastive Learning under Heterophily." _arXiv preprint arXiv:2303.06344_ (2023).
>
> [3] Liu, Nian, et al. "Revisiting graph contrastive learning from the perspective of graph spectrum." _Advances in Neural Information Processing Systems_ 35 (2022): 2972-2983.
>
> [4] Ghose, Amur, et al. "Spectral Augmentations for Graph Contrastive Learning." _International Conference on Artificial Intelligence and Statistics_. PMLR, 2023.
>
> [5] Xie, Xuanting, et al. "Contrastive graph clustering with adaptive filter." _Expert Systems with Applications_ 219 (2023): 119645.
>
> [6] HaoChen, Jeff Z., et al. "Provable guarantees for self-supervised deep learning with spectral contrastive loss." _Advances in Neural Information Processing Systems_ 34 (2021): 5000-5011.
>
> [7] Zhang, Hengrui, et al. "Localized Contrastive Learning on Graphs." _arXiv preprint arXiv:2212.04604_ (2022).

---

> > ### Comment · Reviewer_aFan · 2023-11-22
> > **Acknowledgments**
> >
> > We thank the authors for the detailed description of the literature, allowing to better position this work within the recent advances.

---

> > > ### Author Response · Authors · 2023-11-23
> > > **Official Comment by Authors**
> > >
> > > Thanks to the reviewer for recognizing our work. We have provided responses to some other questions shown in **Summary of the Updates during Rebuttal**, and we will also include more results following the suggestions in the revised version.
> > > If you have any further questions, we welcome further discussion.

---

### Official Review · Reviewer_Q45Y · 2023-11-03

**Soundness:** 3 good
**Presentation:** 3 good
**Contribution:** 2 fair
**Rating:** 5
**Confidence:** 4

**Summary:**

The paper proposes a spectral domain self-supervised method for graphs based on a linear combination of low-pass and high-pass information while keeping the graph structure intact. Although the idea of the paper is interesting and novel, the evaluation lacks comparison with recent works along the similar direction. Moreover, the paper is missing some key references such as FAGCN (not a self-supervised method but considers high pass and low pass information in a supervised manner)[1], [2] [3]. It lacks explanations concerned with the intuition behind low-pass and high-pass information in the learning process.


Theorem 1 is only for regular graphs for binary classification, how can it be a basis of a much stronger claim of having the high pass information in heterophilic datasets?

Have you considered high-pass encoders for the graphs with large heterophily? What is the performance in such a case of alpha = 0? What about when beta = 0?

What about using NT-Xent loss as used in GraphCL as compared to the one used in this paper? What is the intuition of low and high frequency information in the context of the global embedding $g$?

While reporting the results for graphCL, what exact method did you use? node/edge-drop or a combination of other methods?

[1] Deyu Bo et al. , "Beyond Low-frequency Information in Graph Convolutional Networks", AAAI 2021.
[2] Amur Ghosh et al. , "Spectral Augmentations for Graph Contrastive Learning," AISTATS 2023.
[3] W Yang, B Mirzasoleiman, "Graph Contrastive Learning under Heterophily," 2023 Arxiv.


After response:

I appreciate the efforts authors put to clarify my concerns. Thanks. The authors comment on "k-regular graph" restricting the eigenvalues of the normalized Laplacian to [0,2] was unsatisfactory. In fact, it is true for all type of graphs. Such loose statements are not good. Moreover, I was looking for for more intuitions about low pass and high pass questions and comparing against the mean representation, just putting some numbers in order to validate the usefulness of the proposed work seems a bit naive. It would help the paper if frequency domain examples are exploited in simple settings, not just in terms of accuracy numbers after putting a classification head. This paper had a great idea, but could have been much better..

**Strengths:**

please see summary.

**Weaknesses:**

please see summary.

**Questions:**

Theorem 1 is only for regular graphs for binary classification, how can it be a basis of a much stronger claim of having the high pass information in heterophilic datasets?

Have you considered high-pass encoders for the graphs with large heterophily? What is the performance in such a case of alpha = 0? What about when beta = 0?

What about using NT-Xent loss as used in GraphCL as compared to the one used in this paper? What is the intuition of low and high frequency information in the context of the global embedding $g$?

While reporting the results for graphCL, what exact method did you use? node/edge-drop or a combination of other methods?

---

> ### Author Response · Authors · 2023-11-17
> **Reply to Reviewer Q45Y (1)**
>
> Thanks for your insightful feedbacks!
>
> **Reply to Weakness**
>
> > W1: Although the idea of the paper is interesting and novel, the evaluation lacks comparison with recent works along the similar direction.
>
> A1: Following the advice of reviewers, we include SP-GCL[1] and HLCL[2] for further comparison, which are GCLs that address heterophily. We conduct experiments on both synthetic and real-world datasets. The results are summarized in the following tables.
>
>
> | Methods | cora | citeseer | pubmed | cornell | texas | wisconsin | actor | chameleon | squirrel |
> |---------|------|----------|--------|---------|-------|-----------|-------|-----------|----------|
> | SP-GCL  | 82.99 ± 1.18 | 75.54 ± 1.06 | 85.74 ± 0.21 | 69.41 ± 1.49 | 69.76 ± 1.23 | 69.34 ± 0.77 | 35.92 ± 0.67 | 69.23 ± 1.23 | 53.05 ± 1.05 |
> | HLCL    | 85.53 ± 1.03 | 76.79 ± 0.60 | 85.13 ± 0.18 | 64.00 ± 8.98 | 78.38 ± 5.08 | 79.50 ± 4.50 | 40.56 ± 0.70 | 63.86 ± 1.34 | 44.49 ± 0.68 |
> | PolyGCL | 87.57 ± 0.62 | 79.81 ± 0.85 | 87.15 ± 0.27 | 82.62 ± 3.11 | 88.03 ± 1.80 | 85.50 ± 1.88 | 41.15 ± 0.88 | 71.62 ± 0.96 | 56.49 ± 0.72 |
>
>
> | Methods | $\phi=-1$ | $\phi=-0.75$ | $\phi=-0.5$ | $\phi=-0.25$ | $\phi=0$ | $\phi=0.25$ | $\phi=0.5$ | $\phi=0.75$ | $\phi=1$ |
> |---------|------------|--------------|--------------|---------------|-----------|---------------|-----------|---------------|----------|
> | SP-GCL  | 65.82 ± 1.03 | 73.19 ± 0.88 | 68.37 ± 0.89 | 63.72 ± 0.68 | 59.36 ± 0.98 | 73.01 ± 0.51 | 85.52 ± 0.67 | 94.13 ± 0.38 | 88.22 ± 0.49 |
> | HLCL    | 66.03 ± 0.83 | 67.66 ± 0.59 | 70.62 ± 0.63 | 60.80 ± 0.53 | 58.92 ± 0.87 | 65.80 ± 0.40 | 79.25 ± 0.79 | 97.12 ± 0.82 | 93.07 ± 0.80 |
> | PolyGCL | 98.84 ± 0.17 | 94.23 ± 0.31 | 90.82 ± 0.50 | 75.43 ± 0.68 | 66.51 ± 0.69 | 69.43 ± 0.65 | 88.22 ± 0.72 | 98.09 ± 0.29 | 99.29 ± 0.23 |
>
> | Methods | roman-empire | amazon-ratings | minesweeper | tolokers | questions | arxiv-year |
> |---------|--------------|----------------|---------------|-----------|---------------|--------------|
> | SP-GCL  | 63.17 ± 0.22 | 43.11 ± 0.32 | 81.76 ± 0.61 | 80.73 ± 0.62 | 75.08 ± 0.49 | 42.56 ± 0.12 |
> | HLCL    | 67.75 ± 0.19 | 43.92 ± 0.26 | 79.34 ± 0.59 | 78.99 ± 0.67 | 74.92 ± 0.65 | OOM |
> | PolyGCL | 72.97 ± 0.25 | 44.29 ± 0.43 | 86.11 ± 0.43 | 83.73 ± 0.53 | 75.33 ± 0.67 | 43.07 ± 0.23 |
>
>
> Although claimed to tackle heterophily, these two methods still suffer from performance drop in extreme heterophilic cSBM settings when $\phi$ approaches $-1$. In contrast, PolyGCL consistently holds superior performance over the two new baselines on both synthetic and real-world datasets.
>
> >W2: The paper is missing some key references such as FAGCN (not a self-supervised method but considers high pass and low pass information in a supervised manner).
>
> A2: Thanks for your constructive suggestions.
> - As you mentioned, FAGCN [3] considers high-pass and low-pass information in a supervised manner and achieves performance gain in heterophilic graphs. To the best of our knowledge, PolyGCL are the first to achieve efficient learning of low-pass and high-pass filters via polynomial approximation in a self-supervised setting.
> - There are also GCL methods using spectral augmentations [4, 5]. [4] proposes a set of well-motivated graph transformation operations derived via graph spectral analysis, which are spectral graph cropping and graph frequency components reordering. SpCo [5] studies the necessity of high-frequency information in GCL and learns the optimal augmentation from the spectral view. Compared with [4, 5], PolyGCL does not require specially designed or complex preprocessing steps for spectrum augmentations but achieves the contrast between low-pass and high-pass information by directly optimizing the corresponding decoupled filters.
>
> We will further include these works to conduct a more in-depth analysis compared with PolyGCL in the revised version.

---

> ### Author Response · Authors · 2023-11-17
> **Reply to Reviewer Q45Y (2)**
>
> **Reply to Questions**
> > Q1: Theorem 1 is only for regular graphs for binary classification, how can it be a basis of a much stronger claim of having the high pass information in heterophilic datasets?
>
> A3: Thanks for pointing this out. We claim that Theorem 1 is indeed a special case with $k$-regular and binary classification constraints. However, we can generalize Theorem 1 from two aspects.
> - **The $k$-regular constraint.** In Theorem 1, $k$-regular graph is imposed to meet the need of normalized Laplacian $\mathbf{\tilde{L}}$, which restricts its eigenvalues $\lambda_i$ in $[0,2]$. In fact, it can be proved without changing the original text too much, and the introduction of $k$-regular graph is just for convenience of explanation.
> EvenNet [6] states that the analysis based on SRL could be easily extended to the general case, where we directly utilize the unnormalized Laplacian $\mathbf{L}=\mathbf{D}-\mathbf{A}$ to substitute the normalized Laplacian $\mathbf{\tilde{L}}$. In this case, Lemma 1 in Appendix A.1 can be generalized as Lemma 4 in Appendix E.1. Based on Lemma 4 and the unnormalized $\mathbf{L}$, Theorem 1 can be generalized to Theorem 6 in Appendix E.1 shown as follows. Note that the slight difference between Theorem 1 and Theorem 6 is that the constant coefficients of the low-pass/high-pass linear functions are different ($\frac{c}{2}$ and $\frac{nc}{4m}$ respectively). The detailed proof is presented in Appendix E.1.
>
>
>    **Theorem 6 (generalized version on unnormalized Laplacian $\mathbf{L}$ )** For a binary node classification task on graph $\mathcal{G}$ with $n$ nodes and $m$ edges, we consider the linear bounded filter function and the unnormalized Laplacian $\mathbf{L}$ with $\lambda^\prime \geq 0$, for the low-pass filter $g_{low} = c - \frac{nc}{4m}\lambda^\prime$ and the high-pass filter as $g_{high} = \frac{nc}{4m}\lambda^\prime$, then a linear combination of the low-pass and high-pass filter $g_{joint}=x g_{low} + y g_{high}, x \geq 0, y\geq 0, x+y=1$ achieves a lower expected SRL upper bound than $g_{low}$ in heterophilic settings, that is,  $\mathbb{E}_x[\hat{L}\_{joint}] \leq \mathbb{E}_h[\hat{L}\_{low}]$ for $x\sim U(0,1),h\sim U(\frac{1}{2},1)$, where $\hat{L}$ denotes the upper bound for $L$.
>
>
> - **Binary classification.** As for the binary classification in heterophilic graphs, there is a series of works based on spectral analysis of heterophily that adopt the two-class setting [6, 7, 8, 9]. Besides, the theoretical analysis in [7] states that the analysis of multi-class cases can be simplified via the "One vs Others" reduction, that is, for $C$ class classification task, denoting $\boldsymbol{y}^{\prime}_0=\boldsymbol{y}_0$ and $\boldsymbol{y}^{\prime}_1=\sum\_{l=1}^{C-1}\boldsymbol{y}_l$, thus we can transform the multi-class cases into binary classification.
>
> Based on the above discussion, Theorem 1 makes reasonable simplifications and has the potential to be extended to a general form of graph Laplacian or multi-class classification cases, which demonstrate the necessity of introducing high-pass information in heterophilic settings.
>
> >Q2: Have you considered high-pass encoders for the graphs with large heterophily? What is the performance in such a case of $\alpha= 0$? What about when $\beta=0$?
>
> A4: We include experiments where $\alpha$ and $\beta$ are set to zero separately as the ablation studies. Specifically, $\alpha=0, \beta=0$ means only the high-pass/low-pass filter is reserved respectively. The results are given in the following table.
>
> | Methods | cora | citeseer | pubmed | cornell | texas | wisconsin | actor | chameleon | squirrel |
> |---------|-------|-----------|---------|----------|--------|------------|--------|------------|-----------|
> | PolyGCL | 87.57 ± 0.62 | 79.81 ± 0.85 | 87.15 ± 0.27 | 82.62 ± 3.11 | 88.03 ± 1.80 | 85.50 ± 1.88 | 41.15 ± 0.88 | 71.62 ± 0.96 | 56.49 ± 0.72 |
> | PolyGCL($\alpha=0$) | 70.67 ± 0.48 | 64.22 ± 0.93 | 76.41 ± 0.35 | 80.16 ± 2.62 | 86.52 ± 1.97 | 82.07 ± 2.75 | 38.28 ± 0.39 | 68.21 ± 1.40 | 52.10 ± 0.80 |
> | PolyGCL($\beta=0$) | 87.65 ± 0.67 | 78.76 ± 0.75 | 86.64 ± 0.17 | 76.23 ± 5.41 | 82.56 ± 2.13 | 68.88 ± 2.50 | 37.36 ± 0.46 | 65.08 ± 1.27 | 48.52 ± 0.71 |
>
> We observe that the results of PolyGCL($\alpha=0$) remain comparable in heterophilic datasets, while PolyGCL($\beta=0$) shows better adaptability to homophilic settings. The results reveal that the high-pass information is indispensable for graphs with large heterophily, and so is the low-pass information for homophilic graphs. The results of this ablation study are further consistent with the low-pass/high-pass preference for homophilic/heterophilic settings in Figure 3 in Section 5.4. Note that PolyGCL achieves better performance when optimized over both the parameters $\alpha, \beta$ in most datasets.

---

> ### Author Response · Authors · 2023-11-17
> **Reply to Reviewer Q45Y (3)**
>
> >Q3: What about using NT-Xent loss as used in GraphCL as compared to the one used in this paper?
>
> A5: Thanks for your valuable comments. We consider directly substituting our optimization loss in Equation 4 (Section 4.2) with the NT-Xent loss used in GraphCL [6]. Besides, the augmentation strategies and other settings are aligned with GraphCL. The results are listed in the following table.
>
> | Methods | cora | citeseer | pubmed | cornell | texas | wisconsin | actor | chameleon | squirrel |
> |---------|-------|-----------|---------|----------|--------|------------|--------|------------|-----------|
> | PolyGCL | 87.57 ± 0.62 | 79.81 ± 0.85 | 87.15 ± 0.27 | 82.62 ± 3.11 | 88.03 ± 1.80 | 85.50 ± 1.88 | 41.15 ± 0.88 | 71.62 ± 0.96 | 56.49 ± 0.72 |
> | PolyGCL(NT-Xent) | 84.40 ± 0.93 | 76.83 ± 0.94 | 82.63 ± 0.30 | 81.48 ± 2.46 | 84.43 ± 2.95 | 81.75 ± 3.50 | 38.95 ± 0.81 | 69.17 ± 0.94 | 53.30 ± 1.30 |
>
> We observe that the results of PolyGCL(NT-Xent) are slightly lower than PolyGCL. This phenomenon can be attributed to the introduction of structural augmentations (e.g., edge-dropping or subgraph-sampling) destroys the spectral properties of the original graph, which is not conducive to the learning of spectral filters in PolyGCL. However, PolyGCL(NT-Xent) still holds competitiveness with other baselines across different homophilic and heterophilic datasets in Table 2 (Section 5.3), which reflects the universality and effectiveness of the PolyGCL framework.
>
> >Q4: It lacks explanations concerned with the intuition behind low-pass and high-pass information in the learning process. What is the intuition of low and high frequency information in the context of the global embedding $\mathbf{g}$?
>
> A6: We claim the intuition of low and high-frequency information in the context of the global embedding $\mathbf{g}$ lies in the necessity of performing contrastive learning between the decoupled low-pass and high-pass information in Equation 4.
>
> Considering that the global summary $\mathbf{g}$ is defined as $\mathbf{g}=\textbf{Mean}(\mathbf{Z})$, we obtain the final embedding via linear combination as $\mathbf{Z}=\alpha \mathbf{Z}_L+\beta \mathbf{Z}_H$, where $\alpha, \beta$ are linear coefficients that can be set as learnable parameters.
>
> Based on the above, we conclude that $\mathbf{g}$ mixes the decoupled low-pass and high-pass embedding via the linear combination strategy, thus $\mathbf{g}$ contains the low and high-frequency information at the same time in the global embedding. In the learning process of optimizing Equation 4, PolyGCL actually performs contrastive learning between the decoupled low-pass and high-pass embeddings, which mutually boosts the learning of the two branches, and further ensures the final embeddings contain the filtered low-pass and high-pass information to model the graph across different homophily adaptively.

---

> ### Author Response · Authors · 2023-11-17
> **Reply to Reviewer Q45Y (4)**
>
> >Q5: While reporting the results for graphCL, what exact method did you use? node/edge-drop or a combination of other methods?
>
> A7: Following the settings in GraphCL [6], we utilize four augmentations proposed in the original paper, which are node-dropping, edge perturbation, attribute-masking, and subgraph.
>
> In detail, as for reporting the results for GraphCL, we conduct experiments based on the above four augmentations and choose the one that achieves the best result for each dataset. For instance, we use subgraph-sampling for cora, citeseer, cornell, texas, wisconsin, chameleon and squirrel, besides, node-dropping and attribute-masking are chosen for pubmed and actor, respectively.
>
> Note that there is also a hyperparameter $p$ to control the percentage of dropping during augmentations, and we conduct a grid search in $\{0.1, 0.2, ..., 0.9\}$ to obtain the optimal $p$. All these details will be put into the appendix later.
>
> References:
>
> [1] Wang, Haonan, et al. "Can Single-Pass Contrastive Learning Work for Both Homophilic and Heterophilic Graph?." _arXiv preprint arXiv:2211.10890_ (2022).
>
> [2] Yang, Wenhan, and Baharan Mirzasoleiman. "Contrastive Learning under Heterophily." _arXiv preprint arXiv:2303.06344_ (2023).
>
> [3] Bo, Deyu, et al. "Beyond low-frequency information in graph convolutional networks." _Proceedings of the AAAI Conference on Artificial Intelligence_. Vol. 35. No. 5. 2021.
>
> [4] Ghose, Amur, et al. "Spectral Augmentations for Graph Contrastive Learning." _International Conference on Artificial Intelligence and Statistics_. PMLR, 2023.
>
> [5] Liu, Nian, et al. "Revisiting graph contrastive learning from the perspective of graph spectrum." _Advances in Neural Information Processing Systems_ 35 (2022): 2972-2983.
>
> [6] Lei, Runlin, et al. "Evennet: Ignoring odd-hop neighbors improves robustness of graph neural networks." _Advances in Neural Information Processing Systems_ 35 (2022): 4694-4706.
>
> [7] Chen, Zhixian, Tengfei Ma, and Yang Wang. "When Does A Spectral Graph Neural Network Fail in Node Classification?." _arXiv preprint arXiv:2202.07902_ (2022).
>
> [8] Ma, Yao, et al. "Is homophily a necessity for graph neural networks?." _arXiv preprint arXiv:2106.06134_ (2021).
>
> [9] Luan, Sitao, et al. "When do graph neural networks help with node classification: Investigating the homophily principle on node distinguishability." _arXiv preprint arXiv:2304.14274_ (2023).
>
> [10] You, Yuning, et al. "Graph contrastive learning with augmentations." _Advances in neural information processing systems_ 33 (2020): 5812-5823.

---

> ### Author Response · Authors · 2023-11-22
> **Discussion period ending soon; We would like to hear back from Reviewer Q45Y**
>
> Dear Reviewer Q45Y,
>
> We gratefully appreciate your time in reviewing our paper. Since the discussion period ends soon, we sincerely hope our rebuttal has carefully addressed your comments point-by-point. In particular, we positioned our work within some key related works and compared PolyGCL with additional baselines, as suggested by you. Besides, We generalized Theorem 1 to avoid the aforementioned limitations and clarified the intuition behind low-pass and high-pass information in the learning process. Additional experimental results and ablations are also provided to conduct an in-depth analysis of PolyGCL. If you have any other comments or questions, we will be glad to answer them and continue the conversation.
>
> Thank you for your time and attention to this matter.
>
> Best regards,
>
> The Authors of Submission 7529

---

> ### Author Response · Authors · 2023-11-23
> **A Kind Reminder: the author-reviewer discussion period is coming to an end.**
>
> Dear reviewer Q45Y,
>
> We sincerely appreciate your thorough and detailed reviews of our submission. We hope that you will find our responses satisfactory and that they help clarify your concerns. We appreciate the opportunity to engage with you. We would like to kindly remind you that the discussion period is coming to an end. Could you please inform us whether our responses have addressed your concerns or if there are any other questions you need us to clarify?
>
> Thank you very much for your time.
>
> Best regards,
>
> Authors of Submission 7529

---

### Author Response · Authors · 2023-11-20
**Summary of the Updates during Rebuttal**

We sincerely thank all the reviewers for their constructive suggestions and detailed assessments of our paper. We have corrected the spelling and grammatical errors of our paper and posted a revised version based on the reviews, where the additional theoretical analysis and experimental results are supplemented in **Appendix E Updates during Rebuttal**. Here, we would like to summarize our updates during the rebuttal period as follows.

**In the Reply to reviewer Q45Y:**

-   We position our work within some key related works and include two new baselines to make further comparisons with recent works along a similar direction on multiple datasets, which verifies the superiority of PolyGCL in heterophilic settings.
-  We generalize Theorem 1 to avoid the limitations of the regular graph and discuss the strategy to reduce the multi-class cases into binary classification, the detailed theoretical result and proof are shown in Appendix E.1.
- We provide experiment results for the ablation study of $\alpha=0$ or $\beta=0$ to reflect the necessity of high-pass information for heterophilic graphs, and provide results of using NT-Xent loss to show the importance of keeping graph structure intact during augmentation.
- We discuss the intuition behind low-pass and high-pass information in the learning process by further analyzing the context of the global embedding $\mathbf{g}$ in Equation 4 and report the specific augmentation techniques for GraphCL.

**In the Reply to reviewer aFan:**

-   We position our work and its contributions with respect to the papers that the reviewers mentioned and correct the spelling and grammatical errors in our paper.

**In the Reply to reviewer 2heL:**

-   We conduct a deeper analysis of the reason why these particular reparameterization techniques were chosen in PolyGCL based on the experimental comparison between PolyGCL and PolyGCL without reparameterization. In addition, we further discuss the generality and flexibility of the reparameterization by applying it to the Bernstein and Monomial bases.
-   We summarize the reasons for choosing Chebyshev polynomials as the base polynomials theoretically and experimentally. Further, we discuss the potential challenges of filter decoupling, the limitations of PolyGCL, and future work as well.

**In the Reply to reviewer LQgR:**

-   We provide the specific derivation process of Equation (4) to address the concerns of the reviewer about how $\alpha$ and $\beta$ are optimized.
-  We conduct comparative experiments and provide the visualization of the learned filters on cSBM datasets between PolyGCL and directly applying GCL on spectral GNNs, which demonstrates the effectiveness of the learning paradigm in PolyGCL.
-  We included results from additional ablations to conduct an in-depth analysis of PolyGCL. For example, when $\alpha$ and $\beta$ are fixed, or consider other perturbation methods such as edge dropping or subgraph sampling.

We will carefully go through the writings of our paper. If you have any other questions, we are welcome for further discussion.

Thanks for your time and efforts in reviewing.

---

### Meta-Review · Area_Chair_Erz1 · 2023-11-30

**Metareview:**

The paper presents POLYGCL, a Graph Contrastive Learning (GCL) approach addressing challenges in homophilic and heterophilic graphs. Traditional low-pass Graph Neural Network (GNN) encoders introduce smoothness limitations, mitigated by POLYGCL's learnable spectral polynomial filters. Balancing low-pass and high-pass views, it enhances representation learning for both graph types, with theoretical proofs supporting its effectiveness. Empirical evaluations on synthetic and real-world datasets demonstrate POLYGCL's superiority over existing GCL methods, filling a crucial gap in graph representation learning. Specifically focusing on self-supervised learning for spectral GNNs, the authors introduce PolyGCL, a GCL pipeline that incorporates high-pass information, proving effective in addressing contrastive learning challenges in high-pass heterophilic graphs.

The paper addresses a significant problem and presents an intriguing approach. The majority of reviewers express enthusiasm for accepting the paper. Consequently, I also vote in favor of acceptance.

**Justification For Why Not Higher Score:**

The majority of reviewers express satisfaction with accepting this paper, and I also acknowledge its merit for acceptance. However, a reviewer highlighted recent work, such as FAGCN (a supervised method), utilizing high-pass and low-pass information. The authors only partially address this concern, raising the possibility that existing methods may perform comparably well in self-supervised learning. Consequently, I am hesitant to recommend this work for an oral presentation.

**Justification For Why Not Lower Score:**

N/A

---

### Decision · Program_Chairs · 2024-01-16

Accept (spotlight)